# Neural Representational Consistency Emerges from Probabilistic Neural-Behavioral Representation Alignment

Yu Zhu [1 2 3 4]   Chunfeng Song [3]   Wanli Ouyang [3]   Shan Yu [1 2]   Tiejun Huang [4]

## Abstract

Individual brains exhibit striking structural and physiological heterogeneity, yet neural circuits can generate remarkably consistent functional properties across individuals, an apparent paradox in neuroscience. While recent studies have observed preserved neural representations in motor cortex through manual alignment across subjects, the zero-shot validation of such preservation and its generalization to more cortices remain unexplored. Here we present PNBA (**P**robabilistic **N**eural-**B**ehavioral Representation **A**lignment), a new framework that leverages probabilistic modeling to address hierarchical variability across trials, sessions, and subjects, with generative constraints preventing representation degeneration. By establishing reliable cross-modal representational alignment, PNBA reveals robust preserved neural representations in monkey primary motor cortex (M1) and dorsal premotor cortex (PMd) through zero-shot validation. We further establish similar representational preservation in mouse primary visual cortex (V1), reflecting a general neural basis. These findings resolve the paradox of neural heterogeneity by establishing zero-shot preserved neural representations across cortices and species, enriching neural coding insights and enabling zero-shot behavior decoding.

## 1. Introduction

Individual brains exhibit striking structural and physiological heterogeneity in neuronal responses (Churchland et al.,

---
*This work was done during his internship at Shanghai Artificial Intelligence Laboratory. [1]Institute of Automation, Chinese Academy of Sciences [2]School of Artificial Intelligence, University of Chinese Academy of Sciences [3]Shanghai Artificial Intelligence Laboratory [4]Beijing Academy of Artificial Intelligence. Correspondence to: Chunfeng Song <songchunfeng@pjlab.org.cn>, Shan Yu <shan.yu@nlpr.ia.ac.cn>.

*Proceedings of the 42nd International Conference on Machine Learning*, Vancouver, Canada. PMLR 267, 2025. Copyright 2025 by the author(s).

2010) and connectivity patterns (Kasthuri et al., 2015). Yet, neural circuits consistently generate similar functional properties across individuals, from orientation selectivity in visual cortex (Hubel et al., 1959) to movement encoding in motor areas (Churchland et al., 2012). This functional stability persists both within individual brains (Gallego et al., 2020; Stringer et al., 2021) and across different individuals during matched behaviors (Safaie et al., 2023), presenting an apparent paradox with the observed physiological heterogeneity (Averbeck et al., 2006). Understanding this robust preservation despite biological variability is crucial for advancing neural coding theories (DiCarlo et al., 2012) and developing generalizable brain-computer interfaces (BCIs) (Chaudhary et al., 2016; Sussillo et al., 2016).

Recent advances in large-scale neural recording technologies (Stringer et al., 2019; Siegle et al., 2021) have enabled systematic investigation of functional invariance within diverse cortices. Significant progress has been made in motor cortices, observing preserved low-dimensional neural population representations (Rubin et al., 2019; Gallego et al., 2020; Safaie et al., 2023). However, rigorous validation and broader verification of such representation preservation faces three fundamental challenges. First, and most critically, biological neural systems exhibit hierarchical variability spanning multiple scales, from trial-to-trial fluctuations (Churchland et al., 2010; Renart et al., 2010) to inter-individual differences in population activity (Russo et al., 2018; Gallego et al., 2020). This fundamental variability poses a significant barrier for systematic neural analysis. Second, cortical regions serve fundamentally distinct functions. Visual cortex transforms sensory inputs into structured representations (Walker et al., 2019), while motor cortex converts movement goals into coordinated population dynamics (Churchland et al., 2012). Third, these functional differences require distinct analytical frameworks. Sensory systems utilize **encoding** models mapping stimuli to neural responses (Yamins & DiCarlo, 2016; Walker et al., 2019), while motor systems employ **decoding** models predicting movements from neural activity (Sussillo et al., 2016; Pandarinath et al., 2018). These interrelated challenges make rigorous and generalizable validation of preserved neural representations particularly challenging.

Neural population activity has been consistently shown to operate in low-dimensional latent spaces (Churchland et al., 2012; Russo et al., 2018; Stringer et al., 2019). Leveraging this intrinsic low-dimensional structure, we identify three key principles for addressing these challenges. First, explicit neural-behavioral[1] modeling in low-dimensional spaces serves as the foundation to capture functionally relevant representations with biological interpretability (Yamins & DiCarlo, 2016; Walker et al., 2019). Second, neural representations analysis should be anchored in this neural-behavioral framework to preserve functional interpretability. Third, and most crucially, the framework should address the hierarchical neural response variability that enables zero-shot cross-subject validation. Current methods in low-dimensional analysis, including behavior-guided neural activity dimensionality reduction (e.g., CEBRA (Schneider et al., 2023; Chen et al., 2024)) and neural-behavioral fusion strategies (e.g., pi-VAE (Zhou & Wei, 2020; Gondur et al., 2023)), do not fully address these requirements. They either lack explicit neural-behavioral modeling or yield entangled representations that complicate the validation of preserved neural representations. Moreover, their reliance on post-hoc alignment (Safaie et al., 2023; Schneider et al., 2023) fails to address the fundamental challenge of neural variability, precluding direct validation of representation preservation across individuals.

Here we present PNBA (**P**robabilistic **N**eural-**B**ehavioral Representation **A**lignment), a unified framework for learning aligned neural-behavioral representations that enables zero-shot generalization. PNBA introduces **probabilistic representation modeling** to address neural variability, explicit **neural-behavioral correspondence** through dual-modal representation alignment, and **generative constraints** to prevent degenerate representations. These principled considerations enable PNBA to achieve robust alignment between neural activity and behavior, laying the foundation for further investigating neural representations.

We first validated PNBA's alignment capability using neural recordings from motor and sensory cortices across different species (Safaie et al., 2023; Turishcheva et al., 2024). The framework achieved robust neural-behavioral correspondence in both primate motor areas (M1/PMd) during reaching tasks and mouse visual cortex (V1) during visual stimulation. Building on this reliable alignment, we focused on analyzing neural representations. In motor cortices, zero-shot validation revealed consistent neural representations across trials, sessions, and subjects, extending beyond previous findings requiring manual alignment (Safaie et al., 2023). Furthermore, our analysis of mouse V1 revealed preserved neural representations despite fundamentally dif-

ferent recording modalities (calcium imaging) and behavioral contexts. These emerged observations from multiple cortical regions and species reflect fundamental properties of the neural basis. We finally showcase its practicality through zero-shot V1-guided motion decoding, showing the potential for sensory-guided movement BCIs.

We highlight the main contributions of this work below:

• A new probabilistic representation alignment framework, PNBA, that handles cross-scale neural heterogeneity while preventing degenerated representations.

• Robust neural-behavioral representation alignment within multiple cortical regions (M1, PMd, V1) from different species (primates, mice).

• Demonstration of preserved neural representations across trials, sessions, and subjects in distinct cortical regions and species through zero-shot verification, with practical applications in zero-shot behavior decoding.

Codes are availiable at `https://github.com/zhuyu-cs/PNBA`.

## 2. Preliminaries

To investigate the consistency of neural representations in different cortices and species, we here introduce key experimental paradigms, fundamental challenges, and recent breakthroughs in neuroscience.

**Experimental Paradigms.** Understanding neural processing requires investigating how neural populations transform sensory information into behavioral output (Musall et al., 2019; Bahl & Engert, 2020). Driven by different sub-questions in neuroscience, researchers have focused on studying individual cortical regions, e.g., primary visual cortex (V1) for sensory processing and motor areas including primary motor cortex (M1) and dorsal premotor cortex (PMd) for movement planning and control. Visual cortices studies have extensively utilized mouse models, leveraging their genetic tractability (Madisen et al., 2015) to enable large-scale two-photon calcium imaging in head-fixed preparations (Billeh et al., 2020; Stringer et al., 2021) (Figure 1a). Motor cortices investigations have spanned both mice and non-human primates, combining high-density electrophysiology with well-controlled behavioral paradigms (Churchland et al., 2012; Peters et al., 2014) (e.g., center-out task, Figure 1b). These experimental approaches in different model systems provide rich opportunities for investigating neural coding.

**Hierarchical Neural Variability.** Neural activity exhibits remarkable variability across multiple temporal and spatial scales (Fig. 1c). At the finest scale, trial-to-trial variability ($\epsilon_{trial}$) reflects both intrinsic noise and moment-to-

---

[1]We use "neural-behavioral" to refer to neuronal population activity and its associated behavioral variables (e.g., visual stimuli, movement kinematics).

Experimental Paradigms

Hierarchical Neural Variability

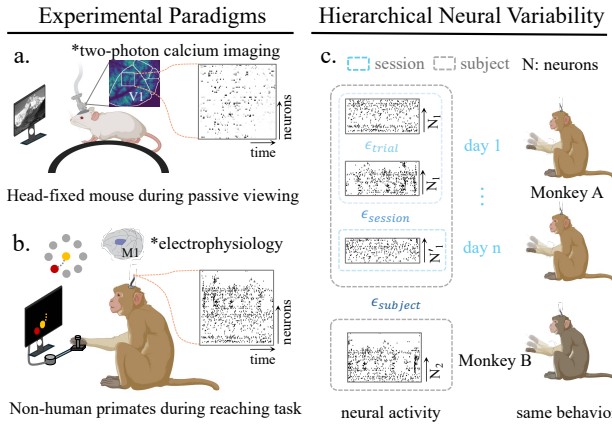

Preserved Neural Representation Across Subjects

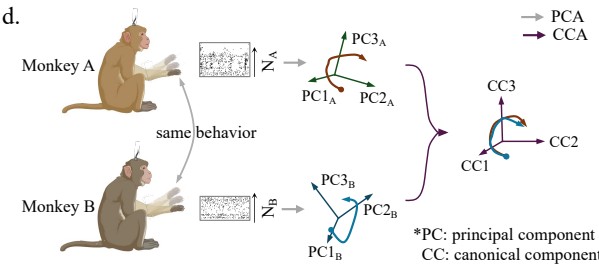

*Figure 1.* **Neural Foundations**. (a) Two-photon calcium imaging in mouse V1 during visual stimulation. (b) Electrophysiological recordings from primate motor cortices during center-out reaching. (c) Hierarchical neural variability across trial ($\epsilon_{\text{trial}}$), session ($\epsilon_{\text{session}}$), and subject ($\epsilon_{\text{subject}}$) scales. (d) Cross-subject alignment pipeline combining PCA-based dimensionality reduction and CCA-based alignment (Safaie et al., 2023).

moment fluctuations in neural responses (Churchland et al., 2010; Ponce-Alvarez et al., 2013). At an intermediate scale, session-to-session variability ($\epsilon_{\text{session}}$) emerges from changes in behavioral state and task-specific adaptation (Peters et al., 2014). The most prominent source of variation occurs at the broadest scale, inter-subject variability ($\epsilon_{\text{subject}}$), stemming from individual differences in neural responses and connectivity patterns. These hierarchical variations lead to a fundamental neuroscience question: how can neural populations maintain reliable function despite such multi-scale variations? Recent advances in population analysis have provided initial insights into this fundamental question.

**Preserved Neural Representations in Motor Cortex.** Despite these complex sources of variability, recent studies have revealed remarkable consistency in neural population activity patterns. At the individual level, reliable trial-to-trial patterns suggest robust underlying mechanisms (Peters et al., 2014; Russo et al., 2018), while across sessions, preserved representations demonstrate stable neural dynamics despite daily variations (Gallego et al., 2020). Most strikingly, in motor cortex, these preservations extend beyond individual subjects (Safaie et al., 2023). Through appropriate dimen-

sionality reduction and alignment, low-dimensional neural activity patterns from different subjects performing identical reaching movements show remarkable similarities (Fig. 1d).

While these findings in motor cortices are compelling, whether such preserved representations exist in other cortical regions like visual cortex remains an open question.

## 3. Method

In this section, we propose a framework for neural-behavior alignment, guided by three principles detailed in Section 1, laying the foundation for neural representation analyses.

### 3.1. Probabilistic Representation Alignment

Given paired observations of neural activities $\mathbf{x} \in \mathcal{X} \subset \mathbb{R}^{N \times T}$ and behavioral variables $\mathbf{y} \in \mathcal{Y} \subset \mathbb{R}^{D \times T}$, where $N$ and $D$ denote the number of recorded neural units and behavioral dimensions respectively, we aim to identify aligned representations that captures neural-behavioral connections. Neural responses exhibit substantial variability across repeated observations while maintaining complex many-to-one mappings with behavior (Marder & Goaillard, 2006; Churchland et al., 2010). This intrinsic variability precludes deterministic point-wise alignments, necessitating a distribution-level framework.

To address this challenge, we formulate our approach through probabilistic representation alignment in a shared latent space $\mathcal{Z}$. Specifically, we construct two encoders: $f_\theta : \mathcal{X} \to \mathcal{P}(\mathcal{Z})$ for neural activities and $g_\phi : \mathcal{Y} \to \mathcal{P}(\mathcal{Z})$ for behavioral variables, mapping inputs to probability distributions (Fig.2a). These distributions are aligned using a probabilistic matching objective (Chun, 2023):

$$\mathcal{L}_{\text{ProbMatch}} = -m \cdot \text{sigmoid}(-a \cdot d(\cdot, \cdot) + b) \\ - (1 - m) \cdot \text{sigmoid}(a \cdot d(\cdot, \cdot) - b) \tag{1}$$

where $m \in \{0, 1\}$ indicates matched or unmatched pairs, with learnable parameters $a$ and $b$ controlling the alignment sensitivity. $d(\cdot, \cdot)$ measures the distributional difference between two multivariate Gaussians:

$$d(f_\theta(\mathbf{x}), g_\phi(\mathbf{y})) = \|\boldsymbol{\mu}_{f_\theta(\mathbf{x})} - \boldsymbol{\mu}_{g_\phi(\mathbf{y})}\|_2^2 + \|\boldsymbol{\sigma}_{f_\theta(\mathbf{x})}^2 + \boldsymbol{\sigma}_{g_\phi(\mathbf{y})}^2\|_1 \tag{2}$$

where $\boldsymbol{\mu}$ and $\boldsymbol{\sigma}^2$ denote the mean and variance of the encoded distributions, respectively.

However, directly optimizing the matching objective leads to degenerate solutions where encoders map diverse inputs into nearly identical representations (analysis in Supplementary Material A.2):

$$f_\theta(\mathbf{x}) \approx g_\phi(\mathbf{y}) \approx \mathbf{z}_{\text{const}}, \quad \forall \mathbf{x} \in \mathcal{X}, \mathbf{y} \in \mathcal{Y} \tag{3}$$

Such degeneration, while achieving alignment, fails to main-

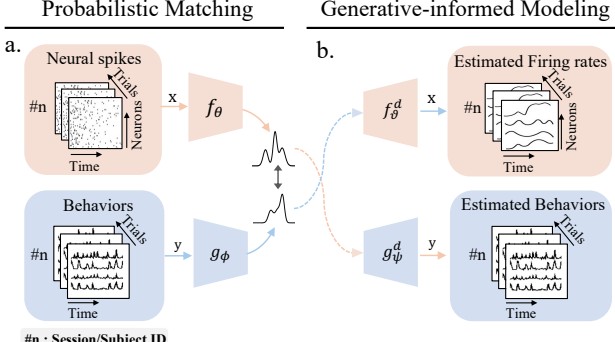

*Figure 2.* **Generative-Informed Probabilistic Framework for Neural-Behavioral Representation Alignment**. The framework comprises (a) a probabilistic matching module with modality-specific encoders $f_\theta$ and $g_\phi$ that project neural activities $\mathbf{x}$ and behavioral variables $\mathbf{y}$ into a shared latent space, and (b) a generative constraint module with decoders $f_\vartheta^d$ and $g_\psi^d$ that preserves modality-specific structure through reconstruction, facilitating robust cross-scale alignment of neural-behavioral representations.

tain meaningful representations, motivating our generative-informed approach to constrain the alignment.

### 3.2. Generative-Informed Representation Alignment

#### 3.2.1. CONSTRAINED OPTIMIZATION FRAMEWORK

To prevent degenerate solutions while maintaining effective alignment, we propose a unified framework that simultaneously ensures distribution matching and structural preservation across modalities. Our key insight is that meaningful alignment requires both distributional matching and generative modeling of each modality (Fig.2b).

Motivated by dimensionality reduction principles in neural data analysis (Stringer & Pachitariu, 2024), we assume that neural-behavioral correlations primarily reside in a shared low-dimensional latent space. This leads to a bidirectional Markov chain ($\mathbf{x} \leftrightarrow \mathbf{z} \leftrightarrow \mathbf{y}$), where neural-behavioral connections are bridged through $\mathbf{z}$. The joint distribution $p(\mathbf{x}, \mathbf{y})$ and conditional distributions $p(\mathbf{y}|\mathbf{x})$ and $p(\mathbf{x}|\mathbf{y})$ together ensure bidirectional mapping preservation.

We formalize this as a constrained optimization problem:

$$\begin{aligned}
\min \; &\mathcal{L}_{\text{ProbMatch}} \\
\text{s.t.} \; &-\log p(\mathbf{x}, \mathbf{y}) \le c_1, \; -\log p(\mathbf{y}|\mathbf{x}) \le c_2, \\
&-\log p(\mathbf{x}|\mathbf{y}) \le c_3
\end{aligned} \quad (4)$$

where $c_1$, $c_2$, and $c_3$ specify upper bounds for negative log-likelihood of joint and conditional distributions. Applying the method of Lagrangian multipliers under the KKT conditions (Karush, 1939), with $\lambda_i \ge 0, c_i > 0$ (Higgins et al.,

2017), we obtain:

$$\begin{aligned}
\mathcal{L}_{\text{total}} = \mathcal{L}_{\text{ProbMatch}} + \lambda_1(-\log p(\mathbf{x}, \mathbf{y})) + \lambda_2(-\log p(\mathbf{x}|\mathbf{y})) \\
+ \lambda_3(-\log p(\mathbf{y}|\mathbf{x}))
\end{aligned} \quad (5)$$

where $\lambda_i$ controls the strength of each generative constraint. Detailed derivations are in Supplementary Material B.1.

Building on the bidirectional Markov chain structure, we factorize the joint distribution in two equivalent forms:

$$p(\mathbf{x}, \mathbf{y}, \mathbf{z}) = \begin{cases} p(\mathbf{x}|\mathbf{z})p(\mathbf{z}|\mathbf{y})p(\mathbf{y}) \\ p(\mathbf{y}|\mathbf{z})p(\mathbf{z}|\mathbf{x})p(\mathbf{x}) \end{cases} \quad (6)$$

This symmetric factorization, combined with the conditional independence assumptions inherent in the Markov structure, leads to a fundamental decomposition of the conditional distributions:

$$\begin{cases} p(\mathbf{x}, \mathbf{z}|\mathbf{y}) = p(\mathbf{x}|\mathbf{z})p(\mathbf{z}|\mathbf{y}) \\ p(\mathbf{y}, \mathbf{z}|\mathbf{x}) = p(\mathbf{y}|\mathbf{z})p(\mathbf{z}|\mathbf{x}) \end{cases} \quad (7)$$

#### 3.2.2. END-TO-END OPTIMIZATION VIA ELBOS

To optimize the intractable generative constraints in Eq.5, we employ variational inference (Kingma & Welling, 2014), approximating the true posteriors $p(\mathbf{z}|\mathbf{x})$ and $p(\mathbf{z}|\mathbf{y})$ with variational distributions $q(\mathbf{z}|\mathbf{x})$ and $q(\mathbf{z}|\mathbf{y})$. The negative log-likelihood terms are optimized by minimizing their respective negative Evidence Lower Bounds (ELBOs):

$$\begin{aligned}
&\lambda_1(-\log p(\mathbf{x}, \mathbf{y})) + \lambda_2(-\log p(\mathbf{x}|\mathbf{y})) + \lambda_3(-\log p(\mathbf{y}|\mathbf{x})) \\
&\le \sum_{i=1}^{3} \lambda_i(-\text{ELBO}_i) \\
&= -\lambda_1(\mathbb{E}_{q(\mathbf{z}|\mathbf{x},\mathbf{y})}[\log p(\mathbf{x}|\mathbf{z})] - KL[q(\mathbf{z}|\mathbf{x},\mathbf{y})||p(\mathbf{z}|\mathbf{y})] \\
&\qquad + \mathbb{E}_{q(\mathbf{z}|\mathbf{x},\mathbf{y})}[\log p(\mathbf{y}|\mathbf{z})] - KL[q(\mathbf{z}|\mathbf{x},\mathbf{y})||p(\mathbf{z}|\mathbf{x})]) \\
&\quad - \lambda_2(\mathbb{E}_{q(\mathbf{z}|\mathbf{y})}[\log p(\mathbf{x}|\mathbf{z})] - KL[q(\mathbf{z}|\mathbf{x})||p(\mathbf{z}|\mathbf{y})]) \\
&\quad - \lambda_3(\mathbb{E}_{q(\mathbf{z}|\mathbf{x})}[\log p(\mathbf{y}|\mathbf{z})] - KL[q(\mathbf{z}|\mathbf{y})||p(\mathbf{z}|\mathbf{x})])
\end{aligned} \quad (8)$$

Following (Johnson et al., 2016), we model the variational posteriors as conditionally independent Gaussian distributions and define the joint approximate posterior as:

$$q(\mathbf{z}|\mathbf{x}, \mathbf{y}) \propto \begin{cases} q_\theta(\mathbf{z}|\mathbf{x})p_\phi(\mathbf{z}|\mathbf{y}), & \text{for} \quad \mathbf{x} \\ q_\phi(\mathbf{z}|\mathbf{y})p_\theta(\mathbf{z}|\mathbf{x}), & \text{for} \quad \mathbf{y} \end{cases} \quad (9)$$

where neural networks with parameters $\theta$ and $\phi$ parameterize the neural activity encoder $f_\theta$ and behavioral encoder $g_\phi$ respectively.

The reconstruction terms in these ELBOs employ modality-specific likelihood functions. Specifically, Poisson negative log-likelihood for neural spikes $\mathbf{x}$ and Gaussian negative

log-likelihood, simplified with mean squared error, for continuous behavioral variables $\mathbf{y}$. These reconstruction objectives are crucial for preventing degenerate representations, as established in the following theorem:

**Theorem 3.1.** *Let $f_\theta : \mathcal{X} \to \mathcal{Z}$ and $g_\phi : \mathcal{Y} \to \mathcal{Z}$ denote the neural and behavioral encoders. The following properties hold:*
*(i) [Non-degeneracy] The optimization objective $\mathcal{L}_{total}$ prevents representation degeneration by ensuring:*

$$\lim_{f_\theta(\mathbf{x}) \to \mathbf{z}_{const}} \mathcal{L}_{total} = +\infty, \quad \forall \mathbf{z}_{const} \in \mathcal{Z} \quad (10)$$

*(ii) [Information Preservation] The generative constraints ensure enhanced mutual information between input and representation spaces:*

$$I(f_{\mathcal{L}_{total}}(\mathbf{x}); \mathbf{x}) \geq I(f_{\mathcal{L}_{ProbMatch}}(\mathbf{x}); \mathbf{x}) + \eta, \quad \eta > 0 \quad (11)$$

*(iii) [Representation Stability] For neural responses $\mathbf{x}_i, \mathbf{x}_j$ corresponding to the same behavioral variable $\mathbf{y}$, the learned representations maintain bounded distances:*

$$0 < \alpha \leq \|f_\theta(\mathbf{x}_i) - f_\theta(\mathbf{x}_j)\|_2 \leq \beta \quad (12)$$

These theoretical guarantees demonstrate that our framework learns representations with non-degenerate mappings of distinct neural patterns, enhanced feature preservation beyond naive distributional matching, and bounded neural variability that maintains behaviorally-relevant structure.

### 3.2.3. CROSS-SUBJECT NETWORK FOR VAIRABLE NEURAL ACTIVTIY

Our approach employs a single shared network across all subjects, despite variations in neural population sizes between recording sessions. This cross-subject architecture processes neural activity matrices ($N \times T$) using shared learnable projection heads at both network endpoints. The input projection transforms data through 2D convolution operations followed by AdaptiveAveragePooling, standardizing the neuron dimension to a fixed size $D$ ($32 \times 8 \times T$ for M1 data and $128 \times 256 \times T$ for V1 data).

This standardization strategy enables the network's encoder, latent representation, and decoder components to be entirely shared across subjects. The output projection mirrors this design symmetrically, employing bilinear interpolation to restore dimensions back to each subject's original neuron count ($N \times T$). By maintaining consistent internal dimensionality while accommodating variable input/output sizes, our architecture achieves true cross-subject sharing of all network parameters, facilitating direct comparison of neural representations across subjects. For detailed implementation specifications, please refer to Supplementary Material C.4.

*Table 1.* **Cross-subject Neural-Behavioral Alignment Performance.** Quantitative evaluation using single-trial Pearson correlation coefficients (R) between neural and behavioral representations across three cortical regions. Higher values indicate better alignment. PNBA demonstrates superior performance across both training and held-out test subjects. Methods marked with ($\S$) require independent training for each modality, while ($\dagger$) denotes approaches necessitating per-trial optimization for individual subjects. Additionally, methods designated with ($*$) require session-specific or subject-specific training procedures.

| Cortical Area | Method | Correlation (R) | |
| --- | --- | --- | --- |
| | | Training Subjects | New Subjects |
| Motor Cortex (M1) | VAE$^\S$ | 0.0197 | 0.0016 |
| | FA+Procrustes† | 0.3334 | 0.2009 |
| | PCA+CCA† | 0.3520 | 0.2160 |
| | FA+amLDS† | 0.5807 | 0.3627 |
| | Neuroformer* | 0.5214 | – |
| | MEME | 0.7756 | 0.7060 |
| | **PNBA (Ours)** | **0.9465** | **0.9302** |
| Motor Cortex (PMd) | VAE$^\S$ | 0.0063 | 0.0028 |
| | FA+Procrustes† | 0.3605 | 0.2877 |
| | PCA+CCA† | 0.3916 | 0.3397 |
| | FA+amLDS† | 0.4733 | 0.4366 |
| | Neuroformer* | 0.3283 | – |
| | MEME | 0.5279 | 0.5255 |
| | **PNBA (Ours)** | **0.9248** | **0.9176** |
| Visual Cortex (V1) | VAE$^\S$ | 0.0029 | -0.0009 |
| | FA+Procrustes† | 0.1221 | 0.1207 |
| | PCA+CCA† | 0.1210 | 0.1209 |
| | FA+amLDS† | 0.1509 | 0.1501 |
| | Neuroformer* | 0.4116 | – |
| | MEME | 0.6357 | 0.5980 |
| | **PNBA (Ours)** | **0.8830** | **0.8705** |

Notably, our PNBA is inherently modality-agnostic, enabling straightforward extension to neural recordings from diverse experimental paradigms, as introduced in Section 2. We validate these theoretical properties through comprehensive experiments with multiple neural recording modalities in the following section.

## 4. Results

Our experimental results first show the robustness of cross-modal representation alignment, then especially reveal preserved neural representations with zero-shot validations, and finally a showcase in V1-driven movement BCIs.

### 4.1. Experimental Settings

**Datasets.** We evaluated our framework on two distinct neural recording datasets to examine cross-modal representation alignment within different neural systems. The primary dataset consists of neural recordings from non-human primates during center-out reaching tasks (Safaie et al., 2023), with recordings from primary motor cortex (M1) and dorsal premotor cortex (PMd). For M1 evaluation, we utilized

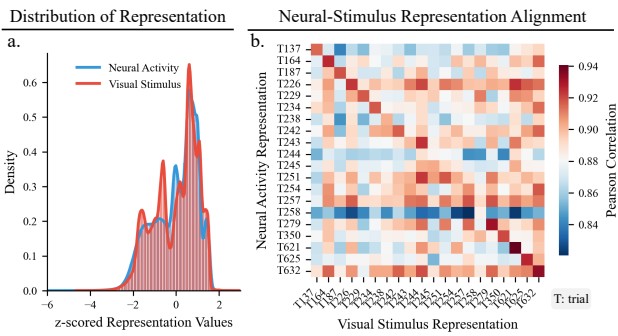

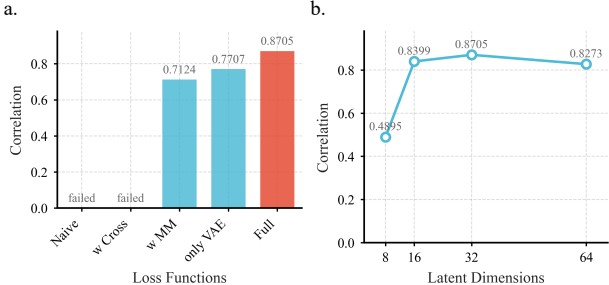

*Figure 3.* **Neural-stimulus representation alignment in mouse V1. a,** Histogram distributions demonstrating aligned representational structure between neural activity and visual stimuli in latent space across two held-out mice. **b,** Trial-wise correlation from the held-out Mouse 9 demonstrating meaningful neural-stimulus correspondence (t = 25.99, p = 1.76 × 10$^{-148}$).

*Figure 4.* **Component analysis and parameter optimization of PNBA framework. a,** Incremental component evaluation starting from baseline probabilistic matching (Naive), incorporating cross-modal VAE (Cross), multi-modal VAE (MM), two VAE-based modeling without matching (only VAE), to the complete PNBA framework. **b,** Neural-behavioral correlation versus latent dimensionality, with optimal performance at d = 32.

neural data from 12 training sessions per subject for two subjects (C, M), two validation sessions per subject, and three test sessions each from two held-out subjects (J, T). Neural spiking activities were recorded concurrently with kinematic trajectories. More details, including PMd, are provided in Supplementary Materials C.1.

To examine the framework's applicability beyond motor cortices, we conducted analyses on calcium imaging data from mouse V1 (Turishcheva et al., 2024). The V1 dataset contains recordings from 10 mice presented with identical dynamic visual stimuli within paired experiments (5 pairs total). We used 8 mice (4 pairs) for training and validation, with 2 mice (1 pair) for held-out testing to assess cross-subject generalization. More details are provided in Supplementary Materials C.1.

**Evaluation Metrics and Baselines.** We employed single-trial Pearson correlation coefficient (R) as the primary evaluation metric, as detailed in SI C.3. This provides a standardized measure of representational similarity within the range [-1, 1]. Given the inherent variability across trials, sessions, and subjects, as well as modality-specific characteristics, correlation values necessarily deviate from the theoretical maximum. We evaluated PNBA against three established baselines, i.e., conventional VAE (Kingma & Welling, 2014) with independent modality training, session-specific Neuroformer (Antoniades et al., 2024) implementing InfoNCE loss (Oord et al., 2018), and MEME (Joy et al., 2021) utilizing mutual distribution supervision. All models were evaluated under consistent training protocols, provided in Supplementary Materials C.2. To assess the statistical significance of the alignment, we conducted independent samples t-tests between matched and mismatched pairs to distinguish task-related neural correspondences from chance-level correlations.

## 4.2. Neural-Behavioral Representation Alignment

**Cross-Modal Alignment Performance.** Quantitative analyses demonstrate that PNBA achieves superior neural-behavioral representation alignment across multiple cortical areas and subjects (Table 1). Using the mouse V1 calcium imaging dataset, we performed systematic validation of alignment quality through distribution-level analyses in new subjects (Figure 3a). The observed distribution overlap between neural activity and behavioral measurements indicates successful encoding of a shared representational space that generalizes across subjects.

Trial-level correlation analyses (Figure 3b) reveal distinct discriminative patterns (t = 25.99, p = 1.76 × 10$^{-148}$), where neural representations exhibit maximal correlation with their corresponding behavioral counterparts while maintaining minimal correlation with non-corresponding pairs. This high-performance alignment result is consistently observed in the monkey motor cortices (Supplementary Figure 9), demonstrating that PNBA effectively captures modality-specific neural-behavioral relationships while maintaining cross-subject generalization. This robust alignment provides a reliable foundation for exploring zero-shot preserved neural representations within different cortices.

**Ablation Studies.** We conducted systematic analyses to evaluate the architectural design of PNBA. Through progressive model comparisons (Figure 4a), we first assessed the contribution of each framework component. Initial experiments with naive probabilistic representation matching yielded suboptimal solutions. Similarly, implementing cross-modal VAE modeling in isolation resulted in degraded representations due to insufficient constraints. The incorporation of multi-modal VAE constraints enabled effective representation alignment (R = 0.71), while exclusive VAE modeling without probabilistic matching achieved R = 0.77. The complete framework, integrating both VAE modeling

and probabilistic matching constraints, demonstrated optimal alignment (R = 0.87), validating the PNBA framework.

We further examined the impact of latent space dimensionality on representation quality. For the V1 dataset, our analyses indicate that a 32-dimensional latent space achieves optimal performance (R = 0.87), as illustrated in Figure 4b. Lower dimensionality proves insufficient for capturing neural-behavioral complexity, while higher dimensions increase computational cost without performance improvement. Following analogous procedures, we determined that 4-dimensional latent representations adequately capture neural dynamics in motor cortical areas (M1 and PMd).

### 4.3. Zero-shot Cross-subject Validation Reveals Hierarchical Preservation in Motor Cortex

Our systematic analysis reveals a hierarchical preservation of neural representations in motor cortex (Fig. 5). Through zero-shot validation on independent held-out subjects, we demonstrate robust preservation across multiple analytical dimensions. The preservation strength exhibits a systematic gradient with increasing temporal and individual variability. Trial-wise patterns demonstrated high consistency (mean R = 0.960 ± 0.011, t-test, p < 0.001), while across-session comparisons revealed substantial stability (R = 0.946 ± 0.008, t-test, p < 0.001). Most importantly, the representations maintained high correlation even across different subjects (R = 0.939 ± 0.033, t-test, p < 0.001), indicating consistent representational organization beyond individual variability (Safaie et al., 2023). Similar preservation patterns were observed in monkey PMd, a region associated with motor planning (see Supplementary Materials D.2).

The robust preservation of neural representations across sessions and subjects reveals a consistent representational structure in motor cortex, despite inherent neural variability. This observation raises a fundamental question regarding whether such preserved representations exist in other cortical areas. To address this question, we next examined the visual cortex, especially V1.

### 4.4. Zero-shot Preserved Neural Representations Extend to Visual Cortex

Zero-shot validation in mouse V1 calcium imaging data revealed that neural representation preservation extends beyond motor cortex. Analysis demonstrated a preservation pattern similar to M1 (Fig. 6). V1 exhibited substantial trial-wise consistency (mean R = 0.912 ± 0.017) (Fig. 11 in Supplementary Material D.3) and, critically, maintained robust cross-subject preservation under zero-shot testing (R = 0.892 ± 0.014, t = 25.38, p = $9.25 \times 10^{-142}$, Fig. 6.c). The preserved representations in V1 remained highly significant across trials and subject.

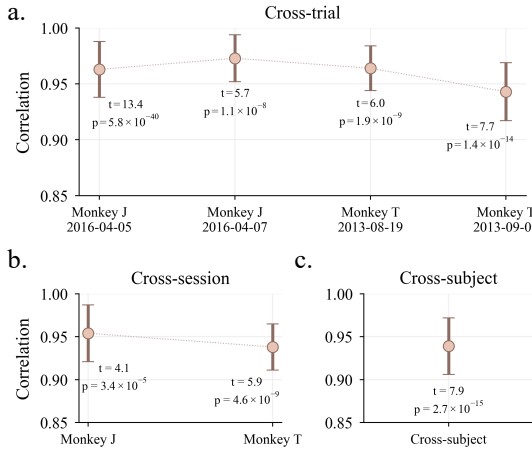

Figure 5. **Hierarchical preservation of neural representations in monkey primary motor cortex (M1).** Correlation analysis demonstrates systematic preservation across trial, session and subject dimensions. **a,** Within-session trial-wise correlations exhibit high consistency (mean R = 0.960 ± 0.011, $n_{J_1}$=196, $n_{J_2}$=198, $n_{T_1}$=135, $n_{T_2}$=208 trials). **b,** Neural representations maintain stability across recording sessions (R = 0.946 ± 0.008). **c,** Zero-shot validation across held-out subjects confirms cross-subject representational similarity (R = 0.939 ± 0.033). Error bars denote standard deviation.

The observation of preserved neural representations across both motor and visual cortices, despite their distinct function, suggests a broader preservation of neural coding structure. This finding has important implications for calibration-free BCIs. In the next section, we demonstrate the practical utility of these preserved representations through a zero-shot V1-guided movement decoding example.

### 4.5. Zero-shot V1-guided Movement Decoding

We leveraged the V1 neural representations to decode mouse movement kinematics, specifically regressing running speed ($\mathbb{R}^2$). The neural representations demonstrated robust cross-subject generalization capabilities across multiple BCI decoders. Using a GRU-based decoder achieved the highest performance ($R^2 = 0.888$, p = $4.44 \times 10^{-13}$), while linear decoders ($R^2 = 0.866$, p = $1.7 \times 10^{-9}$) and feedforward neural networks ($R^2 = 0.880$, p = $1.24 \times 10^{-11}$) also showed promising generalization. These results validate the practicality of the preserved neural representations.

This decoding capability aligns with current understanding of distributed behavioral information in neural circuits, where V1 participates in early sensorimotor integration (Stringer et al., 2019) and shows modulation by behavioral states (Niell & Stryker, 2010). Motor-related signals propagate through various brain regions (Laboratory et al., 2023; Khilkevich et al., 2024), with behavioral information dis-

Preserved Neural Representations Existed in Mouse V1

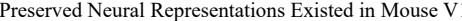

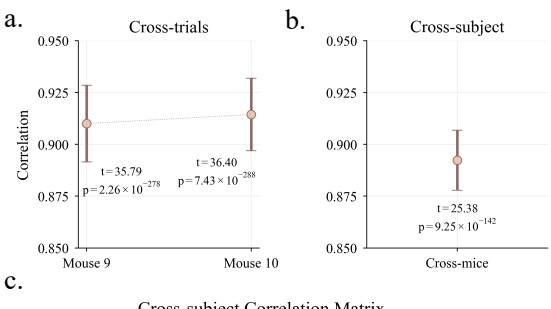

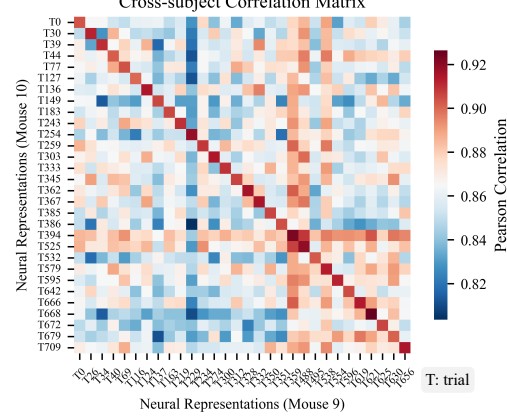

*Figure 6.* **Hierarchical preservation of neural representations in mouse primary visual cortex (V1).** Correlation analysis demonstrates systematic preservation across trial and subject dimensions. **a,** Trial-wise correlations demonstrate high consistency (mean R = $0.912 \pm 0.017$, $n_{M_9} = 412$, $n_{M_{10}} = 413$ trials). **b,** Cross-subject stability analysis between mice presented with identical visual stimuli. **c,** Zero-shot validation confirms cross-subject preservation (R = $0.892 \pm 0.014$). Error bars denote standard deviation.

tributed across neural populations (Musall et al., 2019). Our demonstration of movement decoding provides one example of how V1 preserved neural representations might be leveraged in behavioral decodings, suggesting potential robust motor BCIs that integrate information from multiple cortical regions based on their preserved representations and distributed motor encoding characteristics.

## 5. Related Work

**Neural Representation Learning.** A fundamental analysis strategy in systems neuroscience relies on dimensionality reduction principles to study high-dimensional neural activity, where hidden regularities may emerge despite hierarchical heterogeneity (Stringer & Pachitariu, 2024). Classical dimensionality reduction methods like PCA, jPCA (Churchland et al., 2012), t-SNE (Van der Maaten & Hinton, 2008), and UMAP (McInnes et al., 2018) have revealed such structures but struggle with temporal dynamics. While Transformer models (Ye & Pandarinath, 2021; Liu et al., 2022; Le & Shlizerman, 2022; Ye et al., 2024; Zhang et al.,

Preserved Neural Representations Enables Zero-shot BCI

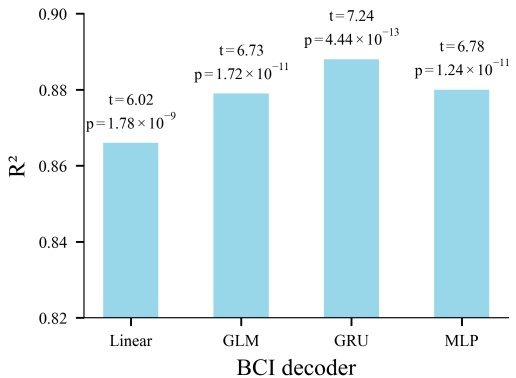

*Figure 7.* **Zero-shot movement decoding from V1 neural activity.** As an application of preserved neural representations, we examined cross-subject movement decoding from V1 recordings. The explained variance ($R^2$) between predicted and actual movement trajectories quantifies decoding performance. Multiple decoding architectures showed significant zero-shot generalization (Linear: t = 6.02, p = $1.78 \times 10^{-9}$; GLM: t = 6.73, p = $1.72 \times 10^{-11}$; GRU: t = 7.24, p = $4.44 \times 10^{-13}$; MLP: t = 6.78, p = $1.24 \times 10^{-11}$).

2024) enhance temporal modeling, their high-dimensional projections conflict with low-dimensional principles and interpretability. Existing solutions using behavior-guided dimensional reduction (Schneider et al., 2023; Chen et al., 2024) and neural-behavioral fusion (Pandarinath et al., 2018; Zhou & Wei, 2020; Keshtkaran et al., 2022; Zhu et al., 2022; Gondur et al., 2023) face fundamental challenges in validating preserved neural representations. Specifically, fusion methods bias representations toward behavioral variables, behavior-guided dimensional reduction approaches fail to establish direct neural-behavioral correspondence, and the required post-hoc alignment may further introduce potential ambiguity. Our PNBA framework addresses these limitations through probabilistic representation alignment, enabling robust low-dimensional representations while naturally bridging neural and behavioral domains.

**Cross-modal Alignment.** Cross-modal representation learning has emerged as a powerful paradigm in machine learning, represented by CLIP (Radford et al., 2021) which demonstrated that dual-encoder architectures with contrastive learning can effectively align image and text representations. This success has inspired numerous advances (Zhai et al., 2023; Chun, 2023; Lavoie et al., 2024). However, aligning neural and behavioral representations presents distinct challenges due to intrinsic neural variability and recording heterogeneity (Fig. 1a,b). Traditional approaches have focused on disentangling behavior-relevant components from neural activity (Sani et al., 2021; Hurwitz et al., 2021; Wang et al., 2024; Sani et al., 2024), but overlooked the importance of behavioral representation learning. Recent methods like Neuroformer (Antoniades et al., 2024) attempt to bridge this gap through contrastive learning, yet

they require subject-specific optimization and assume deterministic neural-behavioral correspondences, failing to account for inherent neural variability. Our probabilistic alignment framework overcomes these limitations by jointly modeling neural variability and behavioral correlation, enabling robust validation of preserved neural representations.

**Neural-Behavioral Modeling.** Neural-behavioral relationships are studied through encoding models mapping stimuli to neural responses and decoding models predicting behavior from neural activity (Mathis et al., 2024). Deep neural networks have enhanced encoding models that mimics the cortices, mapping external stimuli and cognitive variables to neural activity patterns (Yamins & DiCarlo, 2016; Kell et al., 2018; Walker et al., 2019; Bashivan et al., 2019; Marks & Goard, 2021; Vargas et al., 2024), while decoding models extract encoded information (e.g., kinematic variables, visual features) from complex neural activities (Gallego et al., 2017; Yoshida & Ohki, 2020; Stringer et al., 2021). However, these methods assume local neural recordings contain complete behavioral information or require subject-specific fine-tuning. Our PNBA framework addresses these limitations by learning aligned latent representations that capture intrinsic neural-behavioral correlations, maintaining mechanistic interpretability while avoiding direct modeling constraints (Melbaum et al., 2022). The learned representations naturally bridge neural activity and behavioral variables, establishing foundations for BCIs.

## 6. Discussions and Conclusions

While PNBA demonstrates robust implicit cross-subject neural representation alignment for observable behaviors, extending the framework to covert neural processes remains challenging, such as cognitive processes like evidence integration during decision-making or attentional regulation that lack direct behavioral observations. Our PMd analysis provides an intermediate solution by leveraging temporally delayed behavioral readouts, where preparatory neural activity patterns are mapped to subsequent movement kinematics. A promising future direction involves incorporating temporal dynamics modeling (Pandarinath et al., 2018) into the PNBA framework. By explicitly accounting for the dynamical systems properties of neural circuits, such extensions could significantly enhance the current representation alignment approach. We expect that these dynamical enhancements, combined with new experimental paradigms integrated with PNBA, will further advance our understanding of these covert processes.

In this work, we established PNBA as a new framework for robust multimodal representation alignment. Through probabilistic modeling with generative constraints, our approach effectively addresses hierarchical variability across trials, sessions, and subjects while preventing degenerate representations. We validated PNBA through comprehensive zero-shot experiments across multiple cortical regions (M1, PMd, V1) and species (primate, mouse), revealing preserved neural representations that generalize across sessions and individuals. These findings provide new insights into the apparent paradox between neural heterogeneity and functional stability, demonstrating how robust neural representations can emerge and persist despite biological variability, with practical implications demonstrated through zero-shot decoding. In general, we hope PNBA could advance broader neuroscience investigations via multimodal representation alignment, while the discovered zero-shot preserved representations may bring opportunities for stable BCI paradigms, particularly in multi-region BCIs.

## Impact Statement

Our work establishes a computational framework that advances both fundamental neuroscience and neural interface technologies. PNBA's primary contribution is a robust approach for aligning neural and behavioral representations through learned latent spaces. When combined with attribution analysis (Achtibat et al., 2023), this enables precise characterization of neuron-specific encoding mechanisms within neural populations. Beyond single-region analysis, this approach can be extended to broader systems neuroscience investigations, from multi-region to whole-brain analyses, enabling quantitative assessment of area-specific contributions and inter-regional coordination in behavior generation (Steinmetz et al., 2019; Khilkevich et al., 2024).

Furthermore, given that neuroscience inherently relies on multimodal data integration, PNBA's extensible architecture facilitates integration across biological scales, from molecular profiles (Bugeon et al., 2022) to cellular dynamics (Stringer et al., 2019), neural circuit mechanisms (Patriarchi et al., 2020), and ultimately whole-brain analyses (Laboratory et al., 2023).

The preserved neural representations discovered through PNBA present significant potential for advancing calibration-free neural decoding systems, particularly for acute conditions like spinal cord injury (Ahuja et al., 2017) where collecting paired neural-behavioral training data is impractical. Additionally, this framework holds promise for enhancing broader brain-computer interface applications, including speech synthesis (Anumanchipalli et al., 2019; Moses et al., 2021) and motor function restoration (Hochberg et al., 2012).

Collectively, this framework advances our understanding of neural coding while offering promising directions for BCIs, with potential impacts spanning from fundamental neuroscience research to clinical applications.

## Acknowledgement

This work is supported by Shanghai Artificial Intelligence Laboratory.

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

## Supplementary Materials

In the following sections, we provide a comprehensive analysis of cross-modal representation learning approaches for neural-behavioral data integration:

- **Section A. Analysis of Cross-modal Representation Alignment Methods.**

- **Section B: Generatively Informed Neural-Behavior Alignment Framework.** Our novel framework with theoretical guarantees for representation stability and information preservation, including detailed ELBO derivations.

- **Section C: Practical implementation across three neurophysiological datasets (M1, PMd, V1)**, covering dataset organization, network architecture, and training specifications.

- **Section D: Potential related works**, including SwapVAE(Liu et al., 2021), amLDS (Herrero-Vidal et al., 2021) and MARBLE(Gosztolai et al., 2025).

## A. Analysis of Cross-modal Representation Alignment Methods for Neural-Behavioral Alignment

In this section, we analyze cross-modal alignment methods from artificial intelligence that are adapted to neural-behavioral representation alignment. We examine contrastive learning approaches (CLIP (Radford et al., 2021), SigLIP (Zhai et al., 2023) and probabilistic matching (Chun, 2023; Chun et al., 2024)), evaluating their capabilities in handling the inherent variability structure in neural data.

### A.1. Hierarchical Variability in Neural Responses

Neural population recordings exhibit systematic variability at multiple scales (Stringer et al., 2019; Musall et al., 2019), characterized by:

$$0 < \epsilon_{\text{trial}}^{\mathcal{X}} < \epsilon_{\text{session}}^{\mathcal{X}} < \epsilon_{\text{subject}}^{\mathcal{X}} \tag{13}$$

At the **trial level**, variability manifests through firing rate fluctuations within sessions (Churchland et al., 2010; Cohen & Kohn, 2011):

$$\epsilon_{\text{trial}}^{\mathcal{X}} = \sup_{\mathbf{y}} \mathbb{E}_{l_1, l_2} \left[ \|\mathbf{x}_{l_1} - \mathbf{x}_{l_2}\|^2 \right] \tag{14}$$

The **session level** exhibits broader spike count variability patterns (Stringer et al., 2019; 2021):

$$\epsilon_{\text{session}}^{\mathcal{X}} = \sup_{\mathbf{y}} \mathbb{E}_{s_1, s_2} \left[ d_{\mathcal{X}}(P(\mathbf{x}|\mathbf{y}, s_1, i), P(\mathbf{x}|\mathbf{y}, s_2, i)) \right] \tag{15}$$

The largest variation occurs at the **subject level** through inter-individual differences (Russo et al., 2018; Gallego et al., 2020):

$$\epsilon_{\text{subject}}^{\mathcal{X}} = \sup_{\mathbf{y}} \mathbb{E}_{i_1, i_2} \left[ d_{\mathcal{X}}(P(\mathbf{x}|\mathbf{y}, i_1), P(\mathbf{x}|\mathbf{y}, i_2)) \right] \tag{16}$$

This hierarchical structure indicates that neural representations maintain relative consistency within trials and sessions while exhibiting substantial variations across subjects, requiring alignment methods capable of handling multi-scale variability.

### A.2. Analysis of Existing Methods for Neural-Behavior Representation Alignment

We analyze how existing cross-modal alignment methods handle the inherent hierarchical variability in neural data. Neural populations exhibit substantial trial-to-trial variability while maintaining task selectivity (Churchland et al., 2010; Cohen & Kohn, 2011), posing unique challenges for representation alignment.

Existing approaches differ in their treatment of neural variability. CLIP (Radford et al., 2021) employs temperature-scaled contrastive learning:

$$\mathcal{L}_{\text{CLIP}} = -\frac{1}{B} \sum_{i=1}^{B} \log \frac{\exp(\text{sim}(f(\mathbf{x}_i), g(\mathbf{y}_i))/\tau)}{\sum_{j=1}^{B} \exp(\text{sim}(f(\mathbf{x}_i), g(\mathbf{y}_j))/\tau)} - \frac{1}{B} \sum_{i=1}^{B} \log \frac{\exp(\text{sim}(f(\mathbf{x}_i), g(\mathbf{y}_i))/\tau)}{\sum_{j=1}^{B} \exp(\text{sim}(f(\mathbf{x}_j), g(\mathbf{y}_i))/\tau)} \tag{17}$$

SigLIP (Zhai et al., 2023) improved CLIP with sigmoid function:

$$\mathcal{L}_{\text{SigLIP}} = -\frac{1}{B} \sum_{i=1}^{B} \left[ \log \frac{1}{1 + \exp(-\tau \text{sim}(f(\mathbf{x}_i), g(\mathbf{y}_i)) + b)} + \sum_{j \in [B] \setminus \{i\}} \log \frac{1}{1 + \exp(\tau \text{sim}(f(\mathbf{x}_i), g(\mathbf{y}_j)) - b)} \right] \quad (18)$$

Both methods assume direct one-to-one mappings between neural activities and behavioral variables. For neural responses $\{\mathbf{x}_i^k\}_{i=1,k=1}^{M,K}$ associated with behavioral variable $\mathbf{y}_k$, this creates competing objectives:

$$-\|f(\mathbf{x}_i^k) - g(\mathbf{y}_k)\|_2^2 \approx 0 \quad \text{vs.} \quad \exists i, j : \|f(\mathbf{x}_i^k) - f(\mathbf{x}_j^k)\|_2^2 > 0 \quad (19)$$

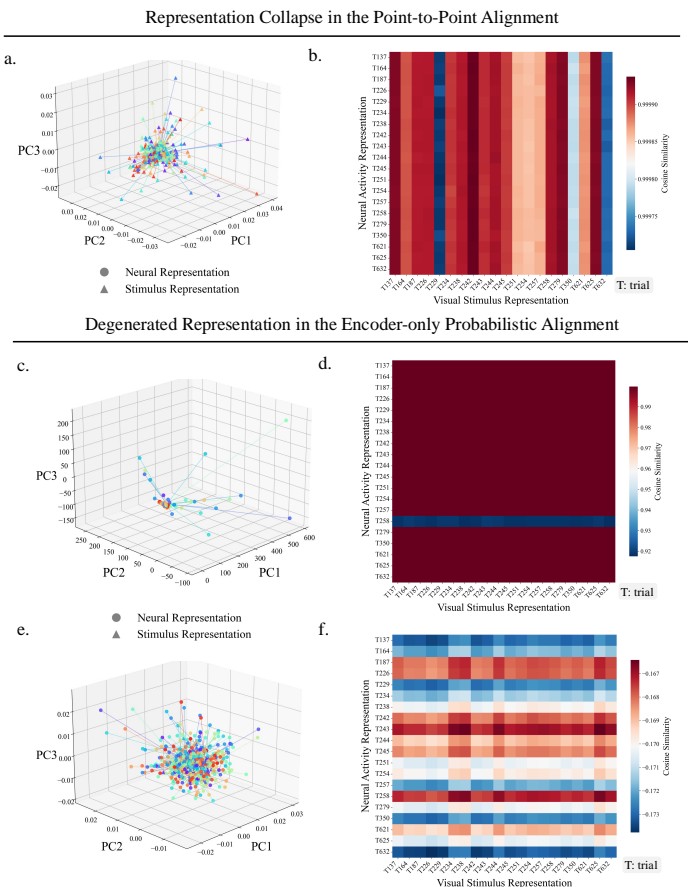

*Figure 8.* **Representation degeneration in naive neural-behavioral alignment methods.** Three-dimensional PCA projections in **a** demonstrate the learned embedding distributions under SigLIP-based alignment, where neural activity encodings $f(\mathbf{x})$ (circles) exhibit severe convergence to a degenerate point-mass distribution while behavioral variable encodings $g(\mathbf{y})$ (triangles) maintain their distributed structure. The trial-wise cosine similarity matrix in **b** shows each entry $(i, j)$ represents $\text{sim}(f(\mathbf{x}_i), g(\mathbf{y}_j))$ for trials T, with statistical analysis (N=825 trials, two mice) revealing no significant discriminative structure (independent samples t-test: t=0.01, p=0.987> 0.05). Under probabilistic matching, PCA visualization in **c** reveals the distributed structure of both neural and behavioral embeddings, with trial-wise similarity matrix in **d** showing consistently high correlation scores ($\text{sim}(f(\mathbf{x}_i), g(\mathbf{y}_j)) \approx 1$) and no statistical difference between matched and mismatched pairs (t=0.02, p=0.981> 0.05). When incorporating information bottleneck regularization ($\lambda = 10^{-4}$), the PCA projection in **e** demonstrates maintained distributional structure but altered geometric relationships, while the corresponding similarity matrix in **f** exhibits systematic negative correlations without achieving significant discrimination (t=0.03, p=0.997> 0.05).

Our analysis (Figure 8a-b) reveals this tension leads to representation degeneration, where neural encoders map different activities to nearly identical embeddings. Such point-to-point correspondence assumptions pose unique challenges for neural datasets, fundamentally distinct from computer vision tasks due to the inherent variability of neural recordings.

Probabilistic matching (Chun, 2023; Chun et al., 2024) attempts to address this by modeling distribution-level alignment:

$$\mathcal{L}_{\text{ProbMatch}} = -\frac{1}{B} \sum_{i=1}^{B} \left[ \log \frac{1}{1 + \exp(-\tau \cdot d(f(\mathbf{x}_i), g(\mathbf{y}_i)) + b)} + \sum_{j \in [B] \setminus \{i\}} \log \frac{1}{1 + \exp(\tau \cdot d(f(\mathbf{x}_i), g(\mathbf{y}_j)) - b)} \right] \quad (20)$$

where $d(f(\mathbf{x}), g(\mathbf{y})) = \|\boldsymbol{\mu}_{f(\mathbf{x})} - \boldsymbol{\mu}_{g(\mathbf{y})}\|_2^2 + \|\boldsymbol{\sigma}_{f(\mathbf{x})}^2 + \boldsymbol{\sigma}_{g(\mathbf{y})}^2\|_1$ However, this formulation admits degenerate solutions where both encoders map to constant values:

$$f'(\mathbf{x}) \approx g'(\mathbf{y}) \approx \mathbf{z}_{\text{const}}, \quad \forall \mathbf{x} \in \mathcal{X}, \mathbf{y} \in \mathcal{Y} \quad (21)$$

Our experiments (Figure 8c-f) confirm that even with information bottleneck regularization (Alemi et al., 2016), simple probabilistic matching remains prone to degenerate solutions, where neural representations become overly similar across distinct behaviors. The analyses demonstrate existing approaches face a fundamental limitation in their inability to simultaneously maintain effective alignment while preserving behavior-wise discriminability. This inherent trade-off between structural preservation and discriminability motivates our development of a generative-informed framework.

## B. Generatively Informed Neural-Behavior Alignment Framework

Neural-behavior alignment requires simultaneously preserving trial-specific neural information and maintaining behavioral consistency. Here, we present a theoretical framework that addresses this challenge through a generative modeling approach. Our framework not only establishes rigorous mathematical guarantees for the alignment process but also reveals key characteristics of preserved neural representations that mirror the hierarchical organization of neural systems.

### B.1. Derivation of the Constrained Optimization Framework

To formalize our neural-behavioral model, we propose a constrained optimization approach that balances probabilistic matching with generative consistency. This appendix provides a detailed derivation of our objective function.

We begin by formalizing our approach as a constrained optimization problem:

$$\begin{aligned} \min \; & \mathcal{L}_{\text{ProbMatch}} \\ \text{s.t.} \; & -\log p(\mathbf{x}, \mathbf{y}) \le c_1 \\ & -\log p(\mathbf{y}|\mathbf{x}) \le c_2 \\ & -\log p(\mathbf{x}|\mathbf{y}) \le c_3 \end{aligned} \quad (22)$$

Here, $\mathcal{L}_{\text{ProbMatch}}$ represents our primary objective function that optimizes the probabilistic matching between neural activity $\mathbf{x}$ and behavior $\mathbf{y}$. The constraints enforce generative consistency by placing upper bounds $c_1$, $c_2$, and $c_3$ on the negative log-likelihood of the joint distribution $p(\mathbf{x}, \mathbf{y})$ and the conditional distributions $p(\mathbf{y}|\mathbf{x})$ and $p(\mathbf{x}|\mathbf{y})$, respectively.

To solve this constrained optimization problem, we apply the method of Lagrangian multipliers under the Karush-Kuhn-Tucker (KKT) conditions (Karush, 1939). For inequality constraints of the form $g_i(\theta) \le c_i$, the KKT conditions require:

1. Stationarity: $\nabla_\theta \mathcal{L}(\theta, \lambda) = 0$

2. Primal feasibility: $g_i(\theta) \le c_i$ for all $i$

3. Dual feasibility: $\lambda_i \ge 0$ for all $i$

4. Complementary slackness: $\lambda_i(g_i(\theta) - c_i) = 0$ for all $i$

In our case, the constraints are:

$$\begin{aligned} g_1(\theta) &= -\log p(\mathbf{x}, \mathbf{y}) \le c_1 \\ g_2(\theta) &= -\log p(\mathbf{y}|\mathbf{x}) \le c_2 \\ g_3(\theta) &= -\log p(\mathbf{x}|\mathbf{y}) \le c_3 \end{aligned} \quad (23)$$

The Lagrangian function is defined as:

$$\mathcal{L}(\theta, \lambda) = \mathcal{L}_{\text{ProbMatch}} + \sum_{i=1}^{3} \lambda_i (g_i(\theta) - c_i) \tag{24}$$

Following the approach in (Higgins et al., 2017), we consider the case where $\lambda_i \geq 0$ and $c_i > 0$. When the constraints are active (i.e., $g_i(\theta) = c_i$), the Lagrangian becomes:

$$\begin{aligned} \mathcal{L}_{\text{total}} &= \mathcal{L}_{\text{ProbMatch}} + \lambda_1 g_1(\theta) + \lambda_2 g_2(\theta) + \lambda_3 g_3(\theta) \\ &= \mathcal{L}_{\text{ProbMatch}} + \lambda_1(-\log p(\mathbf{x}, \mathbf{y})) + \lambda_2(-\log p(\mathbf{y}|\mathbf{x})) + \lambda_3(-\log p(\mathbf{x}|\mathbf{y})) \end{aligned} \tag{25}$$

The Lagrangian multipliers $\lambda_i$ effectively control the strength of each generative constraint, allowing us to balance the probabilistic matching objective with generative requirements.

## B.2. Properties and Guarantees

The framework achieves robust neural-behavior alignment through carefully designed optimization objectives that balance representation similarity with information preservation. We establish the following theoretical guarantees:

**Theorem B.1.** *Let $f_\theta : \mathcal{X} \to \mathcal{Z}$ and $g_\phi : \mathcal{Y} \to \mathcal{Z}$ denote the neural and behavioral encoders. The following theoretical guarantees hold:*

*(i) [Non-degeneracy] The optimization objective $\mathcal{L}_{total}$ prevents representation degeneration by ensuring:*

$$\lim_{f_\theta(\mathbf{x}) \to \mathbf{z}_{const}} \mathcal{L}_{total} = +\infty, \quad \forall \mathbf{z}_{const} \in \mathcal{Z} \tag{26}$$

*(ii) [Information Preservation] The generative constraints ensure enhanced mutual information between input and representation spaces:*

$$I(f_{\mathcal{L}_{total}}(\mathbf{x}); \mathbf{x}) \geq I(f_{\mathcal{L}_{ProbMatch}}(\mathbf{x}); \mathbf{x}) + \eta, \quad \eta > 0 \tag{27}$$

*(iii) [Representation Stability] For neural responses $\mathbf{x}_i, \mathbf{x}_j$ corresponding to the same behavioral variable $\mathbf{y}$, the learned representations maintain bounded distances:*

$$0 < \alpha \leq \|f_\theta(\mathbf{x}_i) - f_\theta(\mathbf{x}_j)\|_2 \leq \beta \tag{28}$$

**Proof of (i):** [Non-degeneracy]

*Proof.* We will proceed by contradiction to establish that any encoder mapping that collapses to a constant representation must yield an unbounded loss value.

Assume there exists a degenerate solution where $f_\theta(\mathbf{x}) = \mathbf{z}\text{const}$ for all $\mathbf{x} \in \mathcal{X}$. Since $\mathcal{X}$ is a compact metric space and $f\theta$ is continuous by construction, this degenerate mapping collapses an uncountable set of distinct neural activities to a single point $\mathbf{z}_{\text{const}}$ in the latent space.

The generative component $\mathcal{L}_{\text{gen}}$ of our total loss involves a probability normalization constraint:

$$\int_{\mathcal{X}} p(\mathbf{x}|\mathbf{z}_{\text{const}})d\mathbf{x} = 1 \tag{29}$$

For this normalization constraint to hold while mapping all points in $\mathcal{X}$ to $\mathbf{z}_{\text{const}}$, the conditional probability density $p(\mathbf{x}_i|\mathbf{z}_{\text{const}})$ for any specific neural activity $\mathbf{x}_i$ must necessarily approach zero:

$$p(\mathbf{x}_i|\mathbf{z}_{\text{const}}) \to 0 \tag{30}$$

This follows from the fact that the probability mass must be distributed across the entire domain $\mathcal{X}$ while maintaining the normalization constraint. Consequently, the log-likelihood term becomes:

$$\log p(\mathbf{x}_i|\mathbf{z}_{\text{const}}) \to -\infty \tag{31}$$

Since our total loss $\mathcal{L}_{\text{total}}$ includes the negative log-likelihood term with weight $\lambda_2 > 0$:

$$\mathcal{L}_{\text{total}} \geq -\lambda_2 \log p(\mathbf{x}|\mathbf{z}) = -\lambda_2 \log p(\mathbf{x}|\mathbf{z}_{\text{const}}) \to +\infty \tag{32}$$

This directly contradicts our objective of minimizing $\mathcal{L}$total. Therefore, for any constant representation $\mathbf{z}_{\text{const}} \in \mathcal{Z}$:

$$\lim_{f_\theta(\mathbf{x}) \to \mathbf{z}_{\text{const}}} \mathcal{L}_{\text{total}} = +\infty \tag{33}$$

$\square$

**Proof of (ii):** [Information Preservation]

*Proof.* We will establish that the encoder optimized with our complete objective $\mathcal{L}_{\text{total}}$ preserves strictly more information than an encoder optimized solely with $\mathcal{L}_{\text{ProbMatch}}$.

Let us decompose the mutual information difference using the entropy formulation:

$$I(f_{\mathcal{L}_{\text{total}}}(\mathbf{x}); \mathbf{x}) - I(f_{\mathcal{L}_{\text{ProbMatch}}}(\mathbf{x}); \mathbf{x}) = [H(\mathbf{x}) - H(\mathbf{x}|f_{\mathcal{L}_{\text{total}}}(\mathbf{x}))] - [H(\mathbf{x}) - H(\mathbf{x}|f_{\mathcal{L}_{\text{ProbMatch}}}(\mathbf{x}))]$$
$$= H(\mathbf{x}|f_{\mathcal{L}_{\text{ProbMatch}}}(\mathbf{x})) - H(\mathbf{x}|f_{\mathcal{L}_{\text{total}}}(\mathbf{x})) \tag{34}$$

The baseline encoder $f_{\mathcal{L}_{\text{ProbMatch}}}$ is optimized solely by the probabilistic matching loss $\mathcal{L}_{\text{ProbMatch}}$, which enforces distributional alignment between neural and behavioral representations. Critically, this objective imposes no explicit constraints on the conditional entropy. The encoder is free to discard information about $\mathbf{x}$ as long as the resulting distribution matches that of the behavioral representations. Thus, there exists a lower bound on the conditional entropy:

$$H(\mathbf{x}|f_{\mathcal{L}_{\text{ProbMatch}}}(\mathbf{x})) \geq \delta_2 \tag{35}$$

In contrast, the encoder $f_{\mathcal{L}_{\text{total}}}$ optimized with our complete objective $\mathcal{L}$total incorporates the evidence lower bound (ELBO) through the generative loss component:

$$\mathbb{E}_{q(\mathbf{z}|\mathbf{x})}[\log p(\mathbf{x}|\mathbf{z})] - KL[q(\mathbf{z}|\mathbf{x})||p(\mathbf{z})] \geq -\delta_1 \tag{36}$$

The reconstruction term $\mathbb{E}q(\mathbf{z}|\mathbf{x})[\log p(\mathbf{x}|\mathbf{z})]$ directly imposes an upper bound on the conditional entropy:

$$H(\mathbf{x}|f_{\mathcal{L}_{\text{total}}}(\mathbf{x})) \leq \delta_1 \tag{37}$$

Since the ELBO constraint explicitly encourages accurate reconstruction through its log-likelihood term, while $\mathcal{L}_{\text{ProbMatch}}$ lacks such reconstruction incentives, we necessarily have:

$$\delta_1 < \delta_2 \tag{38}$$

This establishes the existence of a positive constant $\eta = \delta_2 - \delta_1 > 0$ such that:

$$I(f_{\mathcal{L}_{\text{total}}}(\mathbf{x}); \mathbf{x}) - I(f_{\mathcal{L}_{\text{ProbMatch}}}(\mathbf{x}); \mathbf{x}) \geq \eta > 0 \tag{39}$$

$\square$

**Proof of (iii):** [Representation Stability]

*Proof.* We will establish both upper and lower bounds on the distance between representations of neural activities that correspond to the same behavioral variable.

First, for the upper bound: Since $\mathcal{X}$ is compact and $f_\theta$ is continuous, the image $f_\theta(\mathcal{X})$ is also compact and thus bounded in $\mathcal{Z}$. Let $M = \sup_{\mathbf{x} \in \mathcal{X}} |f_\theta(\mathbf{x})|_2$ denote the maximum norm of any representation. Then, by the triangle inequality, for any $\mathbf{x}_i, \mathbf{x}_j \in \mathcal{X}$:

$$|f_\theta(\mathbf{x}_i) - f_\theta(\mathbf{x}_j)|_2 \leq |f_\theta(\mathbf{x}_i)|_2 + |f_\theta(\mathbf{x}_j)|_2 \leq 2M = \beta \tag{40}$$

For the lower bound, we proceed by contradiction. Suppose no positive lower bound exists, i.e.:

$$\forall \alpha > 0, \exists \mathbf{x}_i, \mathbf{x}_j \in \mathcal{X} \text{ with } \mathbf{x}_i \neq \mathbf{x}_j : |f_\theta(\mathbf{x}_i) - f_\theta(\mathbf{x}_j)|_2 < \alpha \tag{41}$$

This implies the existence of a sequence $(\mathbf{x}_i^{(n)}, \mathbf{x}_j^{(n)})_{n=1}^\infty$ of distinct neural activity pairs corresponding to the same behavioral variable $\mathbf{y}$, such that:

$$\lim_{n \to \infty} |f_\theta(\mathbf{x}_i^{(n)}) - f_\theta(\mathbf{x}_j^{(n)})|_2 = 0 \tag{42}$$

As $n \to \infty$, the representations become arbitrarily close. For any probability model satisfying the normalization constraint:

$$\int_\mathcal{X} p(\mathbf{x}|f_\theta(\mathbf{x}))d\mathbf{x} = 1 \tag{43}$$

The conditional probabilities must satisfy:

$$\lim_{n \to \infty} \left( \log p(\mathbf{x}_i^{(n)}|f_\theta(\mathbf{x}_i^{(n)})) + \log p(\mathbf{x}_j^{(n)}|f_\theta(\mathbf{x}_j^{(n)})) \right) = -\infty \tag{44}$$

This is because as the representations become identical, the probability model must distribute finite probability mass between distinct neural patterns, forcing at least one probability to approach zero. Consequently:

$$\mathcal{L}_{\text{total}} \geq -\lambda_2 \left( \log p(\mathbf{x}_i^{(n)}|f_\theta(\mathbf{x}_i^{(n)})) + \log p(\mathbf{x}_j^{(n)}|f_\theta(\mathbf{x}_j^{(n)})) \right) \to +\infty \text{ as } n \to \infty \tag{45}$$

This contradicts the minimization of $\mathcal{L}_{\text{total}}$. Therefore, there exists $\alpha > 0$ such that for all distinct neural activities $\mathbf{x}_i \neq \mathbf{x}_j$ corresponding to the same behavioral variable $\mathbf{y}$:

$$0 < \alpha \leq |f_\theta(\mathbf{x}_i) - f_\theta(\mathbf{x}_j)|_2 \leq \beta \tag{46}$$

$\square$

*Remark* B.2 (Optimization Balance). The proposed framework achieves representation stability through a balance in the optimization objective. This equilibrium emerges from two complementary mechanisms:

**(1) Clustering Force:** $\mathcal{L}_{\text{ProbMatch}}$ functions as a clustering force, promoting representational similarity among neural responses associated with the same behavioral variables. This ensures behavioral consistency and bounds the maximum distance between related neural representations by $\beta$.

**(2) Regularizing Force:** $\mathcal{L}_{\text{gen}}$ serves as a regularizing force through its generative constraints. By maintaining a minimum distinctiveness threshold $\alpha$, it preserves trial-specific neural information and prevents representation degeneration.

*Remark* B.3 (Key Properties). The theoretical guarantees in Theorem 2 establish three fundamental properties:

**(1) Robustness:** The framework systematically accommodates neural variability while preserving behavior-specific representation structure.

**(2) Information Preservation:** Essential neural response characteristics are retained through rigorous mutual information bounds.

**(3) Representation Stability:** Related neural responses maintain consistent yet distinct representations through provable distance constraints.

*Remark* B.4 (Characteristics of Preserved Neural Representations). Property (iii) reveals the intrinsic characteristics of preserved neural representations: while maintaining maximal similarity ($\leq \beta$), these representations retain inherent distinctiveness ($\geq \alpha$). This mathematical characterization aligns with the hierarchical variability in biological systems, where neural responses exhibit both consistency and inherent variations across different experimental conditions.

## B.3. ELBO Derivation in the Generative modeling.

Building upon the theoretical foundations established above, we now detail the optimization framework that realizes these properties. Our approach leverages a bidirectional generative model structured around two key Markov chains: $\mathbf{x} \to \mathbf{z} \to \mathbf{y}$ and $\mathbf{y} \to \mathbf{z} \to \mathbf{x}$, where $\mathbf{x}$, $\mathbf{y}$, and $\mathbf{z}$ represent neural activities, behavioral variables, and latent representations respectively. This bidirectional structure ensures both neural information preservation and behavioral consistency.

**Behavioral Variable to Neural Activity.** We first derive the ELBO for the forward path, examining how behavioral variables generate corresponding neural activities. This derivation provides two key bounds: one for the joint probability $p(\mathbf{x}, \mathbf{y})$ and another for the conditional probability $p(\mathbf{x}|\mathbf{y})$.

$$
\begin{aligned}
\log p(\mathbf{x}, \mathbf{y}) &= \log \int p(\mathbf{x}, \mathbf{y}, \mathbf{z}) d\mathbf{z} \\
&= \log \int p(\mathbf{x}|\mathbf{z}) p(\mathbf{z}|\mathbf{y}) p(\mathbf{y}) d\mathbf{z} \\
&= \log \int \frac{p(\mathbf{x}|\mathbf{z}) p(\mathbf{z}|\mathbf{y}) p(\mathbf{y})}{q(\mathbf{z}|\mathbf{x}, \mathbf{y})} q(\mathbf{z}|\mathbf{x}, \mathbf{y}) d\mathbf{z} \\
&\geq \mathbb{E}_{q(\mathbf{z}|\mathbf{x},\mathbf{y})} \left[ \log \frac{p(\mathbf{x}|\mathbf{z}) p(\mathbf{z}|\mathbf{y}) p(\mathbf{y})}{q(\mathbf{z}|\mathbf{x}, \mathbf{y})} \right] \\
&= \mathbb{E}_{q(\mathbf{z}|\mathbf{x},\mathbf{y})} [\log p(\mathbf{x}|\mathbf{z})] + \mathbb{E}_{q(\mathbf{z}|\mathbf{x},\mathbf{y})} \left[ \frac{p(\mathbf{z}|\mathbf{y})}{q(\mathbf{z}|\mathbf{x}, \mathbf{y})} \right] + \log p(\mathbf{y}) \\
&= \mathbb{E}_{q(\mathbf{z}|\mathbf{x},\mathbf{y})} [\log p(\mathbf{x}|\mathbf{z})] - KL[q(\mathbf{z}|\mathbf{x}, \mathbf{y})||p(\mathbf{z}|\mathbf{y})] + \log p(\mathbf{y})
\end{aligned}
\tag{47}
$$

where $\log p(\mathbf{y})$ is treated as a constant during optimization since it represents the prior distribution of behavioral variables that is intractable.

Furthermore, we can also have another generation constrain from the conditional probability $p(\mathbf{x}|\mathbf{y})$:

$$
\begin{aligned}
\log p(\mathbf{x}|\mathbf{y}) &= \log \int p(\mathbf{x}, \mathbf{z}|\mathbf{y}) d\mathbf{z} = \log \int p(\mathbf{x}|\mathbf{z}) p(\mathbf{z}|\mathbf{y}) d\mathbf{z} \\
&\approx \log \int p(\mathbf{x}|\mathbf{z}) p(\mathbf{z}|\mathbf{x}) d\mathbf{z} \quad \text{(Since } p(\mathbf{z}|\mathbf{x}) \text{ and } p(\mathbf{z}|\mathbf{y}) \text{ are expected to be highly similar)} \\
&= \log \int p(\mathbf{x}|\mathbf{z}) p(\mathbf{z}|\mathbf{x}) \frac{q(\mathbf{z}|\mathbf{y})}{q(\mathbf{z}|\mathbf{y})} d\mathbf{z} \\
&= \log \mathbb{E}_{q(\mathbf{z}|\mathbf{y})} \left[ p(\mathbf{x}|\mathbf{z}) \frac{p(\mathbf{z}|\mathbf{x})}{q(\mathbf{z}|\mathbf{y})} \right] \\
&\geq \mathbb{E}_{q(\mathbf{z}|\mathbf{y})} [\log p(\mathbf{x}|\mathbf{z})] - KL[q(\mathbf{z}|\mathbf{y})||p(\mathbf{z}|\mathbf{x})]
\end{aligned}
\tag{48}
$$

**Neural Activity to Behavioral Variable** The reverse path analysis considers how neural activities encode behavioral variables, completing our bidirectional framework. Similar to the forward path, we derive bounds for both the joint probability $p(\mathbf{x}, \mathbf{y})$ and the conditional probability $p(\mathbf{y}|\mathbf{x})$.

$$
\begin{aligned}
\log p(\mathbf{x}, \mathbf{y}) &= \log \int p(\mathbf{x}, \mathbf{y}, \mathbf{z}) d\mathbf{z} \\
&= \log \int p(\mathbf{y}|\mathbf{z}) p(\mathbf{z}|\mathbf{x}) p(\mathbf{x}) d\mathbf{z} \\
&= \log \int \frac{p(\mathbf{y}|\mathbf{z}) p(\mathbf{z}|\mathbf{x}) p(\mathbf{x})}{q(\mathbf{z}|\mathbf{x}, \mathbf{y})} q(\mathbf{z}|\mathbf{x}, \mathbf{y}) d\mathbf{z} \\
&\geq \mathbb{E}_{q(\mathbf{z}|\mathbf{x},\mathbf{y})} \left[ \log \frac{p(\mathbf{y}|\mathbf{z}) p(\mathbf{z}|\mathbf{x}) p(\mathbf{x})}{q(\mathbf{z}|\mathbf{x}, \mathbf{y})} \right] \\
&= \mathbb{E}_{q(\mathbf{z}|\mathbf{x},\mathbf{y})} [\log p(\mathbf{y}|\mathbf{z})] + \mathbb{E}_{q(\mathbf{z}|\mathbf{x},\mathbf{y})} \left[ \frac{p(\mathbf{z}|\mathbf{x})}{q(\mathbf{z}|\mathbf{x}, \mathbf{y})} \right] + \log p(\mathbf{x}) \\
&= \mathbb{E}_{q(\mathbf{z}|\mathbf{x},\mathbf{y})} [\log p(\mathbf{y}|\mathbf{z})] - KL[q(\mathbf{z}|\mathbf{x}, \mathbf{y})||p(\mathbf{z}|\mathbf{x})] + \log p(\mathbf{x})
\end{aligned}
\tag{49}
$$

*Table 2.* **Monkey M1 Dataset Organization and Recording Sessions.** The dataset comprises recordings from 4 monkeys performing a center-out (CO) reaching task. Sessions are split into training (24 sessions), validation (4 sessions), and test sets (6 sessions), reflecting the consistent nature of neural responses in this well-established behavioral paradigm.

| Split | Subject | Session Identifiers |
|---|---|---|
| Training | Monkey C | C-CO-20131003, C-CO-20131101, C-CO-20131219, C-CO-20150312, C-CO-20150703, C-CO-20150715, C-CO-20151106, C-CO-20151117, C-CO-20151201, C-CO-20160912, C-CO-20160929, C-CO-20161013 |
| | Monkey M | M-CO-20140203, M-CO-20140307, M-CO-20140626, M-CO-20140929, M-CO-20141203, M-CO-20150512, M-CO-20150610, M-CO-20150615, M-CO-20150617, M-CO-20150623, M-CO-20150625, M-CO-20150626 |
| Validation | Monkey C | C-CO-20131203, C-CO-20160921 |
| | Monkey M | M-CO-20140218, M-CO-20150616 |
| Test | Monkey T | T-CO-20130819, T-CO-20130909 |
| | Monkey J | J-CO-20160405, J-CO-20160407 |

Similarly, in the backward path derivation, $\log p(\mathbf{x})$ represents the prior distribution of neural activities and is also treated as a constant during optimization.

And for the conditional probability $p(\mathbf{y}|\mathbf{x})$:

$$
\begin{aligned}
\log p(\mathbf{y}|\mathbf{x}) &= \log \int p(\mathbf{y}, \mathbf{z}|\mathbf{x})d\mathbf{z} = \log \int p(\mathbf{y}|\mathbf{z})p(\mathbf{z}|\mathbf{x})d\mathbf{z} \\
&\approx \log \int p(\mathbf{y}|\mathbf{z})p(\mathbf{z}|\mathbf{y})d\mathbf{z} \quad \text{(Since } p(\mathbf{z}|\mathbf{x}) \text{ and } p(\mathbf{z}|\mathbf{y}) \text{ are expected to be highly similar)} \\
&= \log \int p(\mathbf{y}|\mathbf{z})p(\mathbf{z}|\mathbf{y})\frac{q(\mathbf{z}|\mathbf{x})}{q(\mathbf{z}|\mathbf{x})}d\mathbf{z} \\
&= \log \mathbb{E}_{q(\mathbf{z}|\mathbf{x})}[p(\mathbf{y}|\mathbf{z})\frac{p(\mathbf{z}|\mathbf{y})}{q(\mathbf{z}|\mathbf{x})}] \\
&\geq \mathbb{E}_{q(\mathbf{z}|\mathbf{x})}[\log p(\mathbf{y}|\mathbf{z})] - KL[q(\mathbf{z}|\mathbf{x})||p(\mathbf{z}|\mathbf{y})]
\end{aligned}
\tag{50}
$$

Note that a fundamental assumption underlying our derivation is the convergence of latent distributions $p(\mathbf{z}|\mathbf{x})$ and $p(\mathbf{z}|\mathbf{y})$ in the learned representation space. This alignment is enforced through probabilistic matching objectives, ensuring that paired neural activities and behavioral variables share similar distributional properties in the latent space. Such alignment is crucial for establishing stable bidirectional mappings between neural and behavioral spaces while preserving their respective information content.

## C. Implementation Details

### C.1. Dataset Description

To comprehensively validate our approach, we conducted experiments on three representative neurophysiological datasets that capture distinct neural encoding paradigms: motor cortex datasets (M1 and PMd) for movement encoding, and a visual cortex dataset (V1) for sensory processing.

**Motor Cortex Datasets (M1 and PMd).** We analyzed neural recordings from primary motor cortex (M1) and dorsal premotor cortex (PMd) of rhesus macaques performing a center-out reaching task, examining how these distinct motor areas encode planned movements. The M1 dataset comprises recordings from 4 monkeys ($\sim$64 neurons/session), while the PMd dataset includes 3 monkeys ($\sim$96 neurons/session), both collected using chronic multielectrode arrays. Neural activity was paired with continuous 2D hand kinematics ($\mathbb{R}^4$: position and velocity). As detailed in Tables 2 and 3, we structured these datasets to enable rigorous evaluation of cross-subject generalization. For M1, we used 24 training sessions and 4 validation sessions from two monkeys (C and M), with 6 test sessions from two held-out monkeys (J and T). The PMd dataset is organized similarly with 20 training sessions and 4 validation sessions from monkeys C and M, and 4 test sessions from monkeys M and T, maintaining parallel evaluation structures across both motor areas while accommodating the available recordings.

*Table 3.* **Monkey PMd Dataset Organization and Recording Sessions.** The dataset comprises recordings from 3 monkeys (C, M, and T) performing a center-out (CO) reaching task. To align with the M1 experimental setup (which uses 4 monkeys), we treat two sessions from Monkey M in the training set as if they were from a separate monkey for zero-shot evaluation purposes. Sessions are split into training (20 sessions), validation (4 sessions), and test sets (4 sessions), maintaining consistency with the M1 experimental paradigm while accommodating the available PMd recordings.

| Split | Subject | Session Identifiers |
|---|---|---|
| Training | Monkey C | C-CO-20160909, C-CO-20160912, C-CO-20160914, C-CO-20160915, C-CO-20160919, C-CO-20160921, C-CO-20160929, C-CO-20161005, C-CO-20161007, C-CO-20161011 |
| | Monkey M | M-CO-20140203, M-CO-20140218, M-CO-20140304, M-CO-20140307, M-CO-20140929, M-CO-20141203, M-CO-20150512, M-CO-20150610, M-CO-20150611, M-CO-20150615 |
| Validation | Monkey C | C-CO-20161013, C-CO-20161021 |
| | Monkey M | M-CO-20150616, M-CO-20150617 |
| Test | Monkey M | M-CO-20150623, M-CO-20150625 |
| | Monkey T | T-CO-20130823, T-CO-20130903 |

*Table 4.* **Dataset Organization and Cross-Mouse Video Stimulus Pairings.** The experimental dataset contains recordings from 10 mice with an 8/2 train-validation split. Each mouse is assigned a unique identifier and systematically paired with another mouse that viewed matching video sequences, enabling direct comparison of neural responses to identical visual stimuli across different subjects.

| Split | Mouse ID | Dataset Identifier | Paired Mouse |
|---|---|---|---|
| Training | Mouse 1 | dynamic29156-11-10-Video-8744edeac3b4d1ce16b680916b5267ce | Mouse 5 |
| | Mouse 2 | dynamic29234-6-9-Video-8744edeac3b4d1ce16b680916b5267ce | Mouse 6 |
| | Mouse 3 | dynamic29513-3-5-Video-8744edeac3b4d1ce16b680916b5267ce | Mouse 7 |
| | Mouse 4 | dynamic29514-2-9-Video-8744edeac3b4d1ce16b680916b5267ce | Mouse 8 |
| | Mouse 5 | dynamic29515-10-12-Video-9b4f6a1a067fe51e15306b9628efea20 | Mouse 1 |
| | Mouse 6 | dynamic29623-4-9-Video-9b4f6a1a067fe51e15306b9628efea20 | Mouse 2 |
| | Mouse 7 | dynamic29647-19-8-Video-9b4f6a1a067fe51e15306b9628efea20 | Mouse 3 |
| | Mouse 8 | dynamic29712-5-9-Video-9b4f6a1a067fe51e15306b9628efea20 | Mouse 4 |
| Validation | Mouse 9 | dynamic29228-2-10-Video-8744edeac3b4d1ce16b680916b5267ce | Mouse 10 |
| | Mouse 10 | dynamic29755-2-8-Video-9b4f6a1a067fe51e15306b9628efea20 | Mouse 9 |

**Visual Cortex Dataset (V1).** We analyzed two-photon calcium imaging data from the primary visual cortex of 10 head-fixed mice viewing naturalistic video stimuli, provided by the Sensorium 2023 Competition. This large-scale dataset captures the activity of 78,853 neurons (∼8,000 neurons/mouse) during passive viewing of 36×64 pixel grayscale video sequences. As shown in Table 4, we employed an 8/2 train-validation split, with a key experimental design feature: each mouse was paired with another subject that viewed identical video sequences. This systematic pairing enables direct comparison of neural representations across individuals while maintaining independent test sets for unbiased evaluation.

These datasets provide complementary perspectives on neural information processing:

- M1 and PMd recordings reveal how different motor areas encode planned movements, with PMd typically showing more complex preparatory dynamics

- V1 recordings demonstrate how sensory information is processed across a large population of neurons, with natural variability in responses across subjects

- All datasets feature carefully structured train/validation/test splits that enable rigorous assessment of cross-subject generalization

## C.2. Training Details

The neural characteristics and computational requirements of motor cortex recordings (M1, PMd) and visual cortex data (V1) required distinct training protocols. Motor cortex datasets feature relatively sparse neural populations paired with

*Table 5.* **Training configurations on three neural datasets**. Our framework utilizes distinct optimization strategies for monkey motor cortex (M1 and PMd) and mouse visual cortex (V1) tasks, with hyperparameters and computational resources tailored to each dataset's characteristics.

| Monkey Motor Cortex (M1 and PMd) | |
|---|---|
| *Optimization Settings* | |
| Optimizer | AdamW ($\beta_1 = 0.9$, $\beta_2 = 0.999$) |
| Weight Decay | 0.05 |
| Batch Configuration | 32 samples, 16 timebins per batch |
| *Learning Rate Schedule* | |
| Initial Learning Rate | $1 \times 10^{-4}$ |
| Warm-up Period | First 600 iterations |
| Peak LR Duration | 50 epochs |
| Decay Strategy | Cosine annealing to $1 \times 10^{-7}$ (last 25 epochs) |
| Total Epochs | 100 |
| *Data & Computing Specifications* | |
| GPU Configuration | $1\times$ NVIDIA A100 (40GB) |
| Training Duration | $\sim$1 hour |
| Inference Latency | 0.3ms |
| Neuron Number | $\sim$64 neurons for M1, $\sim$96 neurons for PMd |
| Behavioral Features | 2D position and 2D velocity ($\mathbb{R}^4$) |

| Mouse Visual Cortex (V1) | |
|---|---|
| *Optimization Settings* | |
| Optimizer | AdamW ($\beta_1 = 0.9$, $\beta_2 = 0.999$) |
| Weight Decay | 0.05 |
| Batch Configuration | 32 samples, 16 timebins per batch |
| *Learning Rate Schedule* | |
| Initial Learning Rate | $1 \times 10^{-4}$ |
| Warm-up Period | First 600 iterations |
| Peak LR Duration | 200 epochs |
| Decay Strategy | Cosine annealing to $1 \times 10^{-7}$ (last 200 epochs) |
| Total Epochs | 400 |
| *Data & Computing Specifications* | |
| GPU Configuration | $4\times$ NVIDIA A100 (40GB) |
| Training Duration | $\sim$8 hours |
| Inference Latency | 1.4ms |
| Neuron Number | $\sim$8000 neurons |
| Visual Stimulus | gray 36$\times$64 image ($\mathbb{R}^{1\times36\times64}$) |

kinematic features, while visual cortex data comprises dense neuronal recordings coupled with high-dimensional visual inputs. We established comprehensive training configurations to address these dataset-specific demands, with detailed optimization parameters, learning schedules, and computational specifications presented in Table 5.

### C.3. Evaluation Metric

We employ Pearson correlation coefficient to evaluate representation similarity. First, we measure cross-modal representation correlation. For neural activity representation $\mathbf{z}_x$ and behavioral representation $\mathbf{z}_y$, both having identical dimensions $d \times T$, the Pearson correlation coefficient is computed as follows:

$$r(\mathbf{z}_x, \mathbf{z}_y) = \frac{\sum_{i=1}^{d \times T}(z_{x,i} - \bar{z}_x)(z_{y,i} - \bar{z}_y)}{\sqrt{\sum_{i=1}^{d \times T}(z_{x,i} - \bar{z}_x)^2 \sum_{i=1}^{d \times T}(z_{y,i} - \bar{z}_y)^2}} \tag{51}$$

where $z_{x,i}$ and $z_{y,i}$ denote the $i$-th element of the flattened representations $\mathbf{z}_x$ and $\mathbf{z}_y$, respectively, and $\bar{z}_x$ and $\bar{z}_y$ represent their respective means.

Additionally, we explore within-modality representational similarity by calculating Pearson correlation coefficients between neural activity representations $\mathbf{z}_x^i$ and $\mathbf{z}_x^j$ from different trials. It is important to note that during model training, we utilized fixed time windows across all trials, preserving complete temporal information for model input. This time-window approach ensured that no information was lost during the training phase.

For evaluation purposes, however, we needed to address the variability in temporal dimension $T$ across trials specifically in the PMd and M1 datasets. In these cases, we aligned the time axes by uniformly subsampling the sequences to the minimum length among the compared trials. This subsampling procedure was not necessary for the mouse V1 dataset, which exhibited consistent temporal dimensions across trials.

After obtaining representations of matching dimensions where required, we computed the Pearson correlation coefficient using the same formula:

$$r(\mathbf{z}_x^{i'}, \mathbf{z}_x^{j'}) = \frac{\sum_{k=1}^{d \times T_{min}}(z_{x,k}^{i'} - \bar{z}_x^{i'})(z_{x,k}^{j'} - \bar{z}_x^{j'})}{\sqrt{\sum_{k=1}^{d \times T_{min}}(z_{x,k}^{i'} - \bar{z}_x^{i'})^2 \sum_{k=1}^{d \times T_{min}}(z_{x,k}^{j'} - \bar{z}_x^{j'})^2}} \tag{52}$$

We determined that selectively resampling at the representation level during evaluation was preferable to full sequence truncation, as it minimized information loss and reduced potential error propagation. This selective approach to temporal

*Table 6.* **Network Architecture Specifications.** Detailed hyperparameters for Monkey M1, PMd and Mouse V1 implementations. The architecture maintains temporal coherence while processing different input modalities.

| Hyperparameters | Monkey M1/PMd | Mouse V1 |
|---|---|---|
| *Projection Head Specifications* | | |
| Input Channels | 1 | 1 |
| Output Channels | 32 | 128 |
| Pooled Dimension ($D$) | 8 | 256 |
| *Encoder Specifications* | | |
| Base Channels (ch) | 32 | 64 |
| Channel Multipliers | (1,2,2,4) | (1,2,2,4) |
| Resolution Levels | 4 | 4 |
| Residual Blocks per Level | 2 | 2 |
| Attention Resolutions | [2,16] | [8,16] |
| Dropout Rate | 0. | 0. |
| *Data Dimensions* | | |
| Neural Activity | $N \times T$ | $N \times T$ |
| Behavior/Stimulus | $4 \times T$ | $36 \times 64 \times T$ |
| *Pre-defined **Low-dimensional** Latent space* | | |
| Latent Space | $1 \times 4 \times T$ | $4 \times 8 \times T$ |
| Reshaped Latent Space | $4 \times T$ | $32 \times T$ |

alignment during evaluation, combined with complete information preservation during training, enabled robust quantification of representational similarities across different experimental conditions.

## C.4. Network Architecture

Our encoder-decoder architecture builds upon UNet, leveraging its proven capability in multi-scale feature extraction while introducing critical modifications for temporal neural data processing. Based on the UNet [2] implementation, we developed a framework specifically optimized for multi-modal neural recordings.

The architecture processes neural activity ($N \times T$) as single-channel temporal sequences, where $N$ represents the number of neurons and $T$ denotes the temporal dimension. To accommodate varying neural population sizes across recording sessions, we implemented learnable projection heads at both network endpoints. The input projection transforms single-channel data through 2D convolution operations (3×3 kernel, stride 1), followed by AdaptiveAveragePooling that standardizes the neuron dimension to a fixed size $D$. This standardization maps the input dimensions from $N \times T$ to $32 \times 8 \times T$ for M1 data and $128 \times 256 \times T$ for V1 data, as detailed in Table 6. The output projection mirrors this structure but employs bilinear interpolation for dimension restoration to $1 \times N \times T$.

The core network comprises a 4-level hierarchical structure with distinct configurations for M1 and V1 datasets. The M1 implementation utilizes 32 base channels, while V1 employs 64 base channels, both following identical channel multiplier patterns. Each hierarchical level incorporates two residual blocks featuring group normalization and timestep embeddings, with self-attention mechanisms strategically placed at specific resolutions to capture long-range dependencies while maintaining computational efficiency.

To ensure consistent representation across modalities, we developed parallel encoders for behavioral and visual data. The behavioral encoder processes M1 kinematic features ($\mathbb{R}^{1 \times 4 \times T}$) by adapting the neural UNet architecture, removing spatial sampling operations while preserving the essential residual blocks and attention mechanisms. For V1 visual stimuli ($\mathbb{R}^{1 \times 36 \times 64 \times T}$), we extended the architecture with 3D convolutions (5×3×3 kernels) to effectively process spatiotemporal data, carefully compressing the stimulus representation to match the neural encoder's latent space.

The architecture converges all modalities into aligned feature spaces through reshaping operations, transforming the representations from $c \times d \times T$ to $(c \times d) \times T$. This unified approach yields compact latent representations of $4 \times T$ for M1 and $32 \times T$ for V1 data, as specified in Table 6. This design significantly reduces computational complexity while preserving the temporal dimension throughout all processing stages, enabling robust capture of temporal dependencies and

---

[2]https://github.com/hojonathanho/diffusion/blob/master/diffusion_tf/models/unet.py

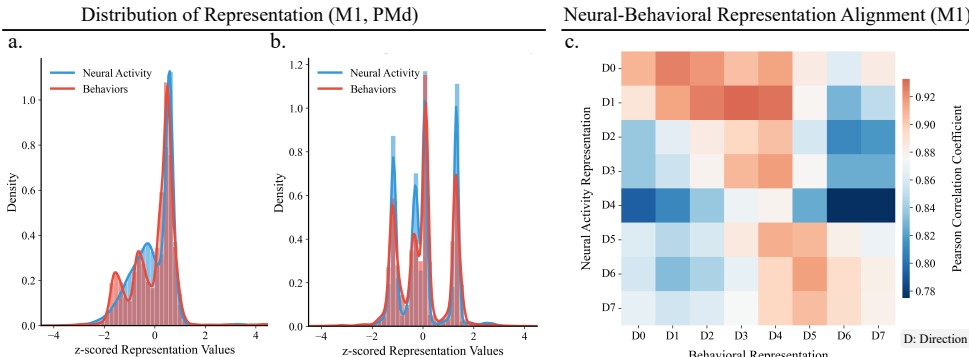

*Figure 9.* **Neural-behavioral representation alignment in motor cortices. a,** Distribution analysis showing aligned representational characteristics between neural activities and behavioral measurements in PMd during center-out reaching. **b,** Corresponding distribution analysis in M1 demonstrating similar alignment properties. **c,** Trial-wise correlation analysis in M1 revealing distributed similarity structure across movement directions (t = 2.17, p = $3.4 \times 10^{-2}$), characteristic of continuous reaching movements.

population dynamics for neural data analysis.

## C.5. Implementation Details for Verifying Preserved Neural Representations

Verification of preserved neural representations necessitates rigorous identification of matched behavioral conditions across recording sessions. For motor cortical regions (M1, PMd), we quantified behavioral similarity using Pearson correlations between kinematic trajectories, with a correlation threshold of R > 0.9 defining matched trials. Visual cortical (V1) analysis leveraged the deterministic nature of stimulus presentations, where identical visual sequences directly established matched conditions. Statistical validation of preserved representations was performed using independent t-tests comparing neural activity correlations between matched and non-matched behavioral conditions.

# D. Extended Results of Cross-modal Alignment and Neural Representation Analysis

Neural representation analysis across scales provides deeper insights into preserved neural representations. Here we present more analyses complementing the main results.

## D.1. Neural-Behavioral Representation Alignment Results of Monkey M1 and PMd

We present comprehensive multi-modal alignment results from monkey motor cortical regions (M1 and PMd) during center-out reaching tasks (Figure 9). Distribution analysis demonstrates that PNBA effectively captures aligned representational properties between neural activities and behavioral measurements in both areas. The overlapping distributions in PMd and M1 indicate successful alignment of neural population dynamics with behavioral kinematics, despite their distinct functional roles in motor control.

The correlation structure in M1 reveals a continuous representational pattern across movement directions (t = 2.17, p = $3.4 \times 10^{-2}$). This distributed similarity is intrinsic to center-out reaching movements (Georgopoulos et al., 1982), where adjacent directional movements share substantial kinematic features, particularly during movement initiation (Churchland et al., 2012). The gradual transitions in neural and behavioral representations across movement directions align with established findings of continuous rotational dynamics in motor cortical populations (Churchland et al., 2012; Russo et al., 2018).

## D.2. Additional Preserved Neural Representation Analyses on Monkey PMd

Following our investigation of preserved neural representations in M1, we conducted a systematic analysis of representational stability in PMd across multiple organizational scales. This hierarchical analysis is particularly crucial for PMd given its central role in motor planning and preparation, where stable representations are essential for consistent movement execution.

Our analysis revealed robust preservation of neural representations at three distinct levels. At the finest scale, within-session trial-to-trial correlation analysis demonstrated high representational stability (mean R = 0.856 ± 0.012, $n_{M_1}$=109, $n_{M_2}$=95,

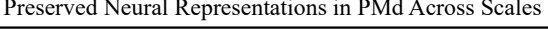

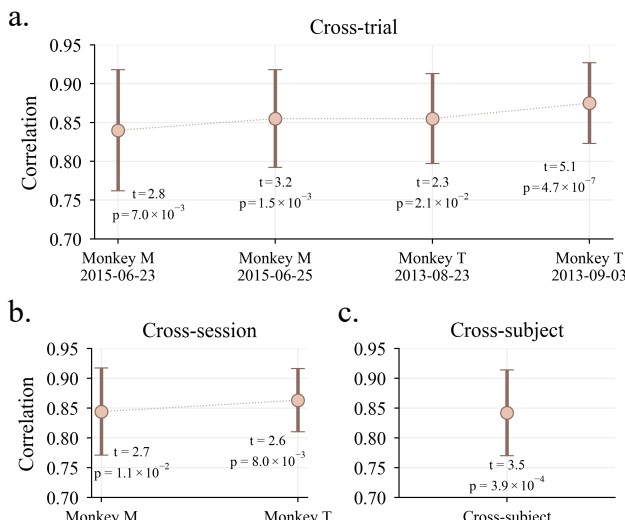

*Figure 10.* **Hierarchical organization of preserved neural representations in monkey dorsal premotor cortex (PMd).** Neural representation similarity analysis demonstrates systematic preservation across organizational levels. **a,** Within-session trial-to-trial correlations exhibit high consistency (mean R = 0.856 ± 0.012, $n_{M_1}$=109, $n_{M_2}$=95, $n_{T_1}$=116, $n_{T_2}$=87 trials). **b,** Neural representations maintain stability between recording sessions (R = 0.854 ± 0.009). **c,** Zero-shot generalization analysis on held-out subjects validates cross-subject representation similarity (R = 0.856 ± 0.072). Error bars denote standard deviation.

$n_{T_1}$=116, $n_{T_2}$=87 trials), indicating reliable encoding of motor plans across repeated behaviors. At the intermediate level, cross-session analysis showed remarkable consistency (R = 0.854 ± 0.009), suggesting neural representation consistency underlying motor planning across different recording periods. Most critically, zero-shot generalization analysis on held-out subjects revealed significant cross-subject representational similarity (R = 0.856 ± 0.072), demonstrating the preservation of fundamental representational features across different individuals.

### D.3. Additional Preserved Neural Representation Analyses on Mouse V1

To validate the robustness of preserved neural representations in V1, we examined the cross-trial correlation patterns in zero-shot mice that were entirely held out during training. Figure 11a presents the correlation matrix for Mouse 9, demonstrating strong preservation of neural activity patterns across trials without any subject-specific fine-tuning. The distinct block-diagonal structure indicates that PNBA successfully captures and maintains condition-specific neural representations in V1, with high correlations observed between trials of the same condition (self-correlation R = 1.0) and low correlations between trials of different conditions. Statistical analysis using independent samples t-tests between matched and mismatched trial pairs yielded p = 0., providing strong evidence for the significant preservation of neural representational structure at the single-trial level. Similar results were observed in Mouse 10 (Figure 11b), confirming PNBA's consistent ability to generalize to completely unseen subjects. These results complement our main findings by demonstrating that PNBA can effectively capture neural dynamics in novel subjects without any additional training or adaptation.

## E. Potential Related Works

This section addresses additional related works suggested during the review process. While these studies explore neural data analysis, they address fundamentally different research questions than our PNBA framework.

Liu et al. (2021)(Liu et al., 2021) proposed SwapVAE, a self-supervised approach for generating neural activity through data augmentation. SwapVAE operates only within the neural activity domain, employing augmentation, i.e., swap operation, based on trial similarity assumption without any behavioral constraints. In contrast, our PNBA framework explicitly bridges neural and behavioral domains through a CLIP-inspired multi-modal paradigm. PNBA directly aligns neural and behavioral representations via generative constraints, which enables zero-shot generalization to completely unseen subjects with varying neural population sizes—a capability that SwapVAE was not designed to address.

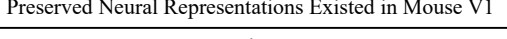

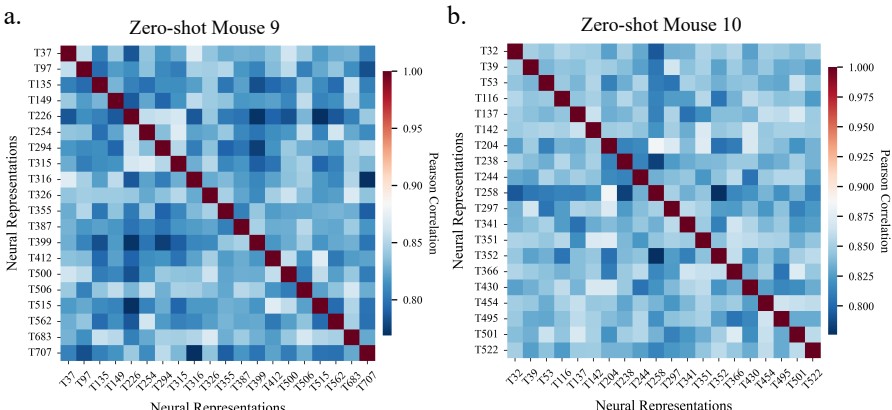

*Figure 11.* **Cross-trial preserved neural representations in mouse primary visual cortex (V1). a.** Correlation matrix for a zero-shot mouse (Mouse 9) reveals robust preservation of neural representational structure, with strong within-condition correlations (diagonal blocks, self-correlation R = 1.0, t=136.715, p=0.) and weak between-condition correlations). Independent samples t-tests between matched and mismatched trial pairs showed highly significant differences (p < 0.001), demonstrating PNBA's ability to maintain distinct neural patterns across experimental conditions without any training data from this subject. **b.** Similar preservation of condition-specific neural representations was observed in another zero-shot mouse (Mouse 10), mean R = 1.0, t=121.494, p=0., highlighting the model's consistent generalization capability to entirely unseen subjects.

Herrero-Vidal et al. (2021) (Herrero-Vidal et al., 2021) developed an "aligned mixture of latent dynamical systems" (amLDS) for cross-animal odor decoding. Their approach explicitly maps neural recordings from different mice into a common latent manifold where neural trajectories are assumed to be similar across animals but distinct across odors. This method fundamentally begins with the assumption that shared neural encoding patterns exist across subjects in response to identical stimuli, and then builds alignment techniques based on this premise. In contrast, PNBA does not presuppose neural encoding similarity across subjects—instead, our approach empirically tests whether such similarities exist by introducing behavioral constraints as the bridging element. While Herrero-Vidal et al. focus on stimulus-driven neural trajectories in olfactory processing, PNBA addresses the broader question of neural-behavioral correspondence during complex, continuous behaviors. Interestingly, our findings provide empirical support for their underlying assumption by demonstrating that shared neural representations do indeed exist under similar behavioral conditions, but our approach arrives at this conclusion without requiring it as a starting assumption.

The recently published MARBLE method (Gosztolai et al., 2025) employs geometric deep learning to obtain interpretable latent representations from neural dynamics. MARBLE decomposes neural dynamics into local flow fields and maps them into a common latent space based on user-defined labels of experimental conditions, allowing similarities between conditions to emerge. While MARBLE provides a similarity metric between dynamical systems and can discover consistent representations across networks and animals, it remains fundamentally a single-modal approach that works exclusively within neural space. In contrast, PNBA implements a distinctly multimodal strategy that directly incorporates behavioral data as constraints in the latent space formation. This fundamental architectural difference—MARBLE's within-neural-domain geometric alignment versus PNBA's cross-modal neural-behavioral alignment—results in approaches optimized for different objectives. MARBLE excels at capturing subtle changes in high-dimensional dynamical flows and relating them to task variables, while PNBA specifically addresses the correspondence between neural patterns and observable behaviors, providing a framework to test whether neural representations preserve their behavioral meaning across subjects.

