# OpenReview forum: "Neural Representational Consistency Emerges from Probabilistic Neural-Behavioral Representation Alignment"
_ICML.cc/2025/Conference — ICML 2025 poster_

### Official Review · Reviewer_DJVs · 2025-02-28

**Overall Recommendation:** 4

**Summary:**

The authors have introduced a new framework PNBA for aligning neural and behavioural distributional representations. Their approach allows learned and generalisable embeddings across subjects. Their framework uses a multimodal VAE architecture with a constrastive loss term (probabilistic matching term) to provide an alignment pressure. It provides a non-linear generalisation to existing alignment tools in neuroscience such as CCA or linear factor models.

**Claims And Evidence:**

The authors have provided a variety of compelling examples on real neural data. Showing cross-trial, -session and -subject correlation analyses in monkey centre out reaching tasks. They also show different neural modality (calcium imaging). Overall empirical evidence is quite strong.

They authors also provide a proof that the generative constraints should prevent mode collapse. They seem quite standard langrangian arguments, but could do with some more detail.

The authors state that the neural representations exhibit minimal correlation with non-corresponding pairs of neural data and behavioural data in figure 3b. But this doesn’t seem so? The correlations between all trials are somewhere between 0.8 and 0.95, including unmatched neural and behavioural data (from my understanding of the plot). This is not convincing.

I don’t see how the method would be ‘present significant potential for advancing calibration-free neural decoding systems’ (line 464). There would still need to be trials from a new animal or session to internally learn the alignment. This would be a form of calibration.

**Essential References Not Discussed:**

A recent paper published just this month in Nature Methods seems to have some relevance for aligning neural representations using geometric deep learning and would not have been available upon submission of this article to ICML: Gosztolai et al. MARBLE.

**Experimental Designs Or Analyses:**

I was not entirely sure about how the correlations are being performed. I assume this is on the latent space z embeddings but I couldn't explicitly find this. This makes it difficult for me to evaluate the soundness of the experimental results/claims.

**Methods And Evaluation Criteria:**

The constrastive probabilistic matching loss seems well motivated given failures of point-wise alignment approaches and the trial-trial variability common in neural data. However, the downside is that the full trial must be input to match with behaviour, hindering online decoding that is important in BCI applications.

However, neural data is inherently dynamic and the VAE approach introduced here does not explicitly model dynamics. Whilst there are a number of methods that do this, i.e., learn embeddings whilst ignoring the dynamics, it prevents forecasting.

**Other Comments Or Suggestions:**

None.

**Other Strengths And Weaknesses:**

I wonder whether there is some combined approach with sequential methods like LFADS or GPFA that could be proposed? Since at the moment it does not explicitly model temporal dynamics.

**Questions For Authors:**

It was unclear to me what the normalised representation values are? Are these the latent space activations? Unit vector normalised? I had to dig into the SI too much to understand what was being correlated in the empirical experiments.

Recent work has shown that there are different ‘solutions’ for solving the same task, i.e., different neural representations. This puts into question a lot of work that uses trial averaging. I wonder how the authors would approach such a problem with PNBA? Since in such a case, I would imagine that the low-dimensional neural representations wouldn’t align?

**Relation To Broader Scientific Literature:**

Cross-subject neural alignment is an open neuroscience problem. Recent work has used linear factor models like CCA or even posthoc alignment. This method provides a clear improvement over these tools, taking advantage of non-linear generative models.

**Theoretical Claims:**

The authors provide short ‘sketch’ proofs for theorems 3.1 and B1. I couldn’t find any glaring errors in the proofs and seem relatively consistent with constrained VAEs of beta VAEs.

---

> ### Author Rebuttal · Authors · 2025-03-28
>
> We sincerely appreciate the reviewer's thoughtful feedback and the positive assessment. We address each point as follows:
>
> ```Q1. The langrangian arguments could do with more detail.```
>
> **A1**: Thanks for this suggestion. In our revision, we will add a more comprehensive derivation of the Lagrangian multiplier method, explicitly clarify how constraints are transformed into optimization objectives, and provide mathematical explanations for key steps in the derivation process.
>
> ```Q2. The authors state that the neural representations exhibit minimal correlation with non-corresponding pairs of neural data and behavioural data in figure 3b. But this doesn’t seem so? The correlations between all trials are somewhere between 0.8 and 0.95, including unmatched neural and behavioural data (from my understanding of the plot).```
>
> **A2**: Thank you for this observation. Correlations in Figure 3b are indeed generally high due to our compact latent space. Our intention was to emphasize trial discriminability, with diagonal elements forming distinguishable patterns in most cases. In our revision, we will better characterize successful cases and emphasize trial-level discriminability.
>
> ```Q3.I don’t see how the method would be ‘present significant potential for advancing calibration-free neural decoding systems’ (line 464). There would still need to be trials from a new animal or session to internally learn the alignment.```
>
> **A3**: PNBA extracts preserved neural representations in zero-shot individuals (Sections 4.3-4.4), allowing us to train both PNBA models and BCI decoders on known individuals and directly apply them to new individuals without any fine-tuning or calibration (Section 4.5). Thus, we believe this demonstrates potential for calibration-free BCI applications.
>
> ```Q4. The downside is that the full trial must be input to match with behaviour, hindering online decoding that is important in BCI applications.```
>
> **A4**: In experiments from Section 4.5, we can adopt a sliding window approach with most BCI decoders, especially Linear and MLP, supporting online decoding. While not our primary focus, this would address the concern. We will add this discussion in our revision.
>
> ```Q5.Unclear how correlations are performed, are they on latent space embeddings?```
>
> **A5**: All correlation calculations are indeed performed on the low-dimensional latent representations z, as they have the same spatial dimensions. We will clarify this explicitly in our revision.
>
> ```Q6.Recently published Gosztolai et al. MARBLE. (2025.3) paper seems relevant.```
>
> **A6**: Thank you for this recommendation. While both MARBLE and our PNBA aim to reduce dimensionality of neural activity, there are fundamental differences. MARBLE is a single-modal method focusing on geometric constraints and temporal dynamics on neural activity, while PNBA is a multimodal strategy directly incorporating behavioral data as constraints in latent space.
>
> We'll add this related work in our revision.
>
> ```Q7. I wonder whether there is some combined approach with sequential methods like LFADS or GPFA that could be proposed? Since at the moment it does not explicitly model temporal dynamics.```
>
> **A7**:We agree and believe such a combination is entirely feasible and would address the current limitation in modeling temporal dynamics. PNBA provides multimodal alignment constraints while methods like LFADS/GPFA offer complementary temporal modeling capabilities. These represent independent constraints that could be directly combined in future work.
>
> We'll include this as limitation and state this future work in our revision.
>
> ```Q8. It was unclear to me what the normalized representation values are? Are these the latent space activations? Unit vector normalized?```
>
> **A8**:'Normalized representation values' refers to z-score normalization applied independently to each trial's representation, making different modalities comparable by eliminating scale differences. We'll clarify this in our revision.
>
> ```Q9. I wonder how the authors would approach such 'multiple solutions' with PNBA? Since in such a case, I would imagine that the low-dimensional neural representations wouldn’t align?```
>
> **A9**: PNBA directly addresses the "multiple solutions" challenge. Rather than forcing identical representation for the same behavior, our approach:
>
> 1. Avoids assuming one-to-one correspondence between neural activity and behavior, instead employing distributional matching that preserves neural diversity.
>
> 2. Ensures that PNBA must generate *different* neural representations for the same behavior (Theorem 1, Property 1), otherwise it would result in representation collapse.
>
> 3. Establishes **a lower bound on representational similarity that ensures differences** and an upper bound to promote feasible similarity (Theorem 1, Property 3).
>
> Therefore, these multiple considerations make PNBA particularly suited for studying the many-to-one mapping between neural activity and behavior.

---

### Official Review · Reviewer_PNyz · 2025-03-14

**Overall Recommendation:** 4

**Summary:**

Modeling shared variability across multiple animals is critical to understand universal principles of neural computation. Still, we like probabilistic tools to capture them into a singular representation. This work introduces a new probabilistic method to represent neural and behavioral variability across animals while allowing for individual variability. The authors validate their model in two different neural datasets, and compared the performance to relevant alternative models showing the applicability of their model.

**Claims And Evidence:**

The work is clearly motivated, supported by the theoretical proof and the presented results. The competitive performance results in comparison to alternative models further cements the relevance of the presented model. Moreover, they tested the model performance across recording modalities and species illustrating the broader applicability of their method. The author claim the relevance to BCI applications, still this requires additional constrains, including exploration of inference times, computational costs and data demands.

**Essential References Not Discussed:**

The authors should include a reference, and potential comparison, to prior work introducing probabilistic method for across-animal task-informed neural alignment (Herrero-Vidal et al. NeurIPS 2021).

**Experimental Designs Or Analyses:**

The authors evaluate their method on two neural datasets which amply shows the applicability of the work. Still, using simulation could further highlight uses and limitations of the model. For example, can the model generalize across missing behavioral conditions? Can the model recover ground truth parameters? How robust is the model to the dimensionality of the latent space, missing observations, trial misalignment, or individual variability? Moreover, adding behavioral decoding results across all the experiments and datasets would further show the ability of the model to extract those representations.

**Methods And Evaluation Criteria:**

The methods are adequate and evaluation as a function of correlation between trajectories captures the shared variability across neural representations. The authors also showed minimal decoding performance to unseen observations. Still, since the emphasis of the work is to align to behavioral tasks, decoding with respect to behavior would provide a better sense of how much behavioral information is captured. To provide additional evidence on the use of the method for BCI, beyond just neuroscience discovery, the authors should discuss limitations on real-time inference and computational demands.

**Other Comments Or Suggestions:**

The authors could include comparisons to linear alignment methods and compare decoding performance, data demands, and computational costs to understand the direct applicability to BCI technology.

**Other Strengths And Weaknesses:**

While the work is clearly presented, adding a section to discuss the limitations is needed to fully assess the impact of the contribution.

**Questions For Authors:**

How much data is needed to train the initial model? The author mention that a linear head is needed to compensate for inconsistencies between the number of recorded neurons, how much of the alignment happens in this transformation? How much data is used for this pre-alignment?

**Relation To Broader Scientific Literature:**

The authors correctly frame their work around the relevant literature, they compared their model to alternative solutions. Still, the authors missed prior work introducing a probabilistic method for across-animal task-informed neural alignment (Herrero-Vidal et al. NeurIPS 2021). Additionally, the author could contrast their results with other simpler alignment methods based on Procrustes alignment (Williams et al NeurIPS 2021, Safaie et al. Nature 2023).

**Theoretical Claims:**

The theoretical proofs are clearly presented with enough detail and are correct.

---

> ### Author Rebuttal · Authors · 2025-03-28
>
> We thank the positive assessment and constructive feedback. We address each point as follows:
>
> ```Q1. Decoding behavior would better demonstrate captured behavioral information. Authors should discuss limitations on real-time inference and computational demands for BCI.```
>
> **A1**: Behavior decoding results are as follows: M1, R²=0.89, PMd, R²=0.83 (training subjects); M1, R²=0.78, PMd, R²=0.71 (testing subjects), indicating capture of behavioral information. These are expected as PNBA aligns neural-behavioral representations. To avoid circular reasoning, Section 4.5 presents movement decoding using V1 data, where the neural encoder was only trained with stimulus in PNBA, excluding movement data.
>
> For BCI discussions, training takes ~1 hour (M1/PMd) and ~8 hours (V1), detailed in SI Line 1075, with inference requiring 0.3ms and 1.4ms, respectively (averaged over 1000 runs on an A100 GPU), showing potential for BCI.
>
> We will add these discusions in our revision.
>
> ```Q2. Simulations could reveal model limitations. Can the model generalize across missing behavioral conditions or recover ground truth parameters?How robust is it to latent space dimensionality, trial misalignment, or individual variability?```
>
> **A2**: Thanks for this suggestion. We agree with the usefulness of simulation. However, building simulations for neural-behavioral modeling with ground truth representation constraints is inherently difficult due to the complex, unknown nature of true neural encoding mechanisms. We therefore validated PNBA across three diverse real datasets.
>
> Regarding specific questions:
>
> a. No,  PNBA cannot generalize to missing behavioral conditions in the raining. This is a current limitation, as discussed in Section 6.
>
> b. PNBA doesn't aim to recover ground truth parameters, as these are difficult to define in real neural data. PNBA is data-driven, learning effective representation alignment instead, similar to CLIP.
>
> c. We show robustness through theoretical guarantees (Theorem 3) and empirical validation. Figure 4b shows the results of varying latent space dimensions. Figures 3b,6c show effective handling of trial-to-trial variability in unseen subjects.
>
> ```Q3. The authors missed a reference, Herrero-Vidal et al. NeurIPS 2021.```
>
> **A3**: Thanks for suggesting this work.
>
> Herrero-Vidal et al. assumes similar neural trajectories across individuals responding to identical stimuli so as to align **single-modal** neural activity through individual-specific optimization. In contrast, our **multimodal** neural-behavioral alignment doesn't presuppose neural encoding similarity, and can be tested across individuals. This methodological difference allows us to validate preserved neural representations in unseen subjects.
>
> We believe our findings provide empirical support for assumptions in Herrero-Vidal et al., supporting shared neural representations under the same behavior.
>
> We will add these discussions in our revision.
>
> ```Q4. Compare with Procrustes alignment methods (Williams et al. 2021, Safaie et al. 2023).```
>
> **A4**: Thanks for this suggestion. We didn't directly compare with Williams et al. and Safaie et al. for several reasons:
>
> a. Our work proposes cross-modal (neural-behavioral) alignment, while these methods focus on unimodal (neural-neural) alignment, making direct comparison potentially unfair.
>
> b. **Our method requires no fine-tuning on unseen individuals, whereas these methods require individual-specific optimization**.
>
> c. Different foundational assumptions (**A3**).
>
> We will include these discussions in our revision.
>
> ```Q5. Add a section discussing limitations.```
>
> **A5**: Limitations are currently discussed in Section 6, paragraph 1, noting PNBA cannot train with neural activity alone. We will add temporal dynamics discussion based on Reviewer ```DJVs```'s suggestion.
>
> ```Q6. Compare with linear alignment methods regarding decoding performance, data demands, and computational costs for BCI applications.```
>
> **A6**: For computational costs and linear alignment comparisons, please see **A1** and **A4**. Regarding data requirements, PNBA is currently trained with 2 monkeys (M1/PMd) or 8 mice (V1), indicating moderate demands. While BCI applications aren't our current focus, our results show potential, and we plan dedicated BCI research in future work.
>
> We will add these discussions in our revision.
>
> ```Q7. How much data is needed for initial training? How does the transformation for handling different neuron counts work, and how much data is used for pre-alignment?```
>
> **A7**: Our model requires no pre-training, but trains directly on data from multiple individuals in the training set and directly tests on new individuals without fine-tuning.
>
> To handle varying neuron counts, we use a **shared** convolutional projection head followed by pooling to only unify dimensions in our network. This requires no separate pre-alignment data or process, as it's learned directly during end-to-end training.

---

> > ### Comment · Reviewer_PNyz · 2025-04-06
> >
> > I thank the authors for the detailed feedback. While the assumptions differ between models, they are still relevant comparisons and would further show that assumptions presented in this work's model are more adequate for the underlaying data statistics. Still, I agree with the additional points of discussion and I adjusted the score accordingly. Please add the relevant comparisons discussed here to the final manuscript.

---

> > > ### Author Response · Authors · 2025-04-07
> > >
> > > We appreciate the reviewer's insightful suggestion! We have now included all suggested baselines. As shown in the following table, FA+Procrustes† and PCA+CCA† achieve moderate performance despite being optimized specifically for each trial of individual subject. The results suggest that linear alignment approaches struggle to effectively handle complex, high-dimensional neural activity and behavioral variables, further highlighting the importance of non-linear algorithms in this domain.
> > >
> > > | Cortical Area | Method | Training Subjects | New Subjects |
> > > |---------------|--------|-------------------|--------------|
> > > | Motor Cortex (M1) | VAE§ | 0.0197 | 0.0016 |
> > > | | FA+Procrustes† | 0.3334 | 0.2009 |
> > > | | PCA+CCA† | 0.3520 | 0.2160 |
> > > | | FA+amLDS† | 0.5807| 0.3627 |
> > > | | Neuroformer* | 0.5214 | -- |
> > > | | MEME | 0.7756 | 0.7060 |
> > > | | **PNBA (Ours)** | **0.9465** | **0.9302** |
> > > | Motor Cortex (PMd) | VAE§ | 0.0063 | 0.0028 |
> > > | | FA+Procrustes† | 0.3605 | 0.2877 |
> > > | | PCA+CCA† | 0.3916 | 0.3397 |
> > > | | FA+amLDS† | 0.4733| 0.4366 |
> > > | | Neuroformer* | 0.3283 | -- |
> > > | | MEME | 0.5279 | 0.5255 |
> > > | | **PNBA (Ours)** | **0.9248** | **0.9176** |
> > > | Visual Cortex (V1) | VAE§ | 0.0029 | -0.0009 |
> > > | | FA+Procrustes† | 0.1221 | 0.1207 |
> > > | | PCA+CCA† | 0.1210 | 0.1209 |
> > > | | FA+amLDS† | 0.1509| 0.1501|
> > > | | Neuroformer* | 0.4116 | -- |
> > > | | MEME | 0.6357 | 0.5980 |
> > > | | **PNBA (Ours)** | **0.8830** | **0.8705** |
> > > §: Independent modality training without cross-modal alignment
> > > *: Requires session-specific/subject-specific training
> > > †: Requires per-trial optimization for each individual subject
> > >
> > > Note: We utilized Factor Analysis (FA) to standardize dimensionality when adapting the amLDS algorithm (Herrero-Vidal et al. 2021) for our neural-behavioral representation alignment task.
> > >
> > > We hope these additional results address the reviewer's suggestions. We also note that these results would satisfy the comparison (**Q2**) suggested by Reviewer ```a8zJ```. We will incorporate all these comparisons into the final manuscript as suggested.

---

### Official Review · Reviewer_ahz6 · 2025-03-17

**Overall Recommendation:** 3

**Summary:**

This work proposed a probabilistic representation alignment framework PNBA that can be used to align neural activities and animal behaviors. The method is applied across brain regions, neural data modalities, and animal species. Authors provided extensive experimental evidence across multiple datasets, validating the robustness of the proposed method.

**Claims And Evidence:**

Line 382-384: "The observation of preserved neural representations across both motor and visual cortices, despite their distinct functional roles and varying correlation strengths, suggests a broader preservation of neural coding structure." This claim is confusing, and seems like an over-claim: How is the calcium imaging encoder trained? Is the zero-shot experiments still the cross-subject zero-shot experiments? Is the behavioral encoder frozen? If both encoders are optimized based on the defined loss given by the authors, the experimental evidence does not suggest broader preservation of neural coding structure.

**Essential References Not Discussed:**

Liu, Ran, Mehdi Azabou, Max Dabagia, Chi-Heng Lin, Mohammad Gheshlaghi Azar, Keith Hengen, Michal Valko, and Eva Dyer. "Drop, swap, and generate: A self-supervised approach for generating neural activity." Advances in neural information processing systems 34 (2021): 10587-10599.

The proposed idea is actually very similar to the work pointed above. This work uses different encoders to encode neural activities of different animals, uses a behavioral-guided latent space, and uses generative loss to prevent latent collapse.

**Experimental Designs Or Analyses:**

This paper provides experimental results on three different datasets involving different modalities and brain regions. The experimental design is extensive, seem to be complete, and are of high-quality.

However, the authors provided limited experimental details in the main text, which makes certain parts difficult to evaluate. For example, one common issue in spike neural data analysis is the difference between amount of neurons across sessions, and how to transfer encoder given different input sizes. How did the author address this issue? Are encoders re-trained when initialized on the data from a new animal? How did the authors deal with unseen neurons in new animals/sessions?

**Methods And Evaluation Criteria:**

1. As stated in Section 3.1, the proposed matching objective between neural activities and behaviors is only one necessary condition for distributional alignment. It seems to me that one sufficient condition for alignment is that f and g both needs to be reversible. Is this true? Yet the provided theoretical guarantees seem to be very loose to provide a sufficient alignment.

2. What is the underlying assumption of using Pearson correlation coefficient to evaluate the alignment quality? (1) Pearson Correlation is a method to evaluate the distributional similarity between two variables. In your case, how it is applied on single-trial data? Is it applied on each coordinate of a latent representation z? Are the latent vectors are centered (i.e. are you measuring cosine similarity)? (2) Have the authors tried other possible evaluation methods for alignment, e.g. L1 distance? Are all baseline methods optimized based on the given evaluation metrics?

**Other Comments Or Suggestions:**

NA

**Other Strengths And Weaknesses:**

NA

**Questions For Authors:**

NA

**Relation To Broader Scientific Literature:**

The problem studied is a very important problem to the neuroscience community.

**Theoretical Claims:**

I did not check carefully, as the theoretical bounds seem to be loose.

---

> ### Author Rebuttal · Authors · 2025-03-28
>
> We sincerely thank the reviewer for the constructive feedback. We address each point as follows:
>
> ```Q1. Line 382-384’s claim is confusing. How is the calcium imaging encoder trained?Is the zero-shot experiments still the cross-subject zero-shot experiments?Is the behavioral encoder frozen?If both encoders are, the evidence does not suggest broader preservation. ```
>
> **A1**: We appreciate the reviewer's careful examination.
>
> a. Our conclusion summarizes preservation properties observed in three independent experiments (M1, PMd, V1) during zero-shot cross-subject conditions. We will clarify this in our revision.
>
> b. The calcium imaging encoder follows the same PNBA framework but with visual stimulus encoders. The zero-shot evaluations are also performed on unseen subjects without any fine-tuning or further alignment process.
>
> c.  During training, all encoders are optimized.
>
> d.  We understand the concern about preservation claims. However, our conclusion is supported by (1) consistent preservation observed in functionally different brain regions (M1, PMd, V1); (2) such validation are performed on unseen subjects without further finetuning or alignment. We believe this rigorous validation provides evidence for the preservation property.
>
> ```Q2. The proposed matching objective in Section 3.1 is only one necessary. Is reversibility of encoders a sufficient condition for alignment?The theoretical guarantees seem too loose for sufficient alignment.```
>
> **A2**: We agree with the reviewer's insight. While reversible encoders would provide an optimal sufficient condition for alignment, this becomes impractical for neural data requiring dimensionality reduction. Our generative constraint approach offers a practical sufficient condition that balances theoretical guarantees with implementation feasibility, achieving effective representation alignment on multiple datasets without requiring strict encoder reversibility.
>
> ```Q3. What assumptions underlie using Pearson correlation for alignment quality?(1)How is it applied to single-trial data?Is it per coordinate?Are vectors centered?(2)Did you try other metrics like L1 distance?Are all baseline methods optimized based on the given evaluation metrics? ```
>
> **A3**: The core assumption behind using Pearson correlation is that aligned neural and behavioral representations should exhibit similar distributional structures in latent space.
>
> (1) We compute correlation between complete neural-behavioral latent vectors for each trial, not coordinate-wise. Pearson correlation centers the vectors and normalizes by standard deviation, distinguishing it from cosine similarity.
>
> (2) We considered alternatives (L1, Euclidean) but selected Pearson correlation because it's established in neuroscience for representational similarity analysis (e.g., Safaie et al., Nature 2023) and effectively captures structural similarities in high-dimensional spaces.
>
> All methods (baselines and our PNBA) were optimized using their original objective functions, not optimized the evaluation metrics.
>
> We will include these details in our revision.
>
> ```Q4. Limited details are provided in the main text regarding how to handle different neuron counts across sessions and how to transfer encoders given different input sizes. Are encoders re-trained when initialized on the data from a new animal? How do you deal with unseen neurons in new animals/sessions? ```
>
> **A4**: Thank you for highlighting this important issue. We addressed varying neuron counts through a neuron-adaptive encoder architecture incorporating projection layers with pooling layer (SI Line 1091-1093) to get unified dimensional latent space. This enables direct application to new animals with different input sizes.
>
> No, encoders are not re-trained when applied to new animals.
>
> Our approach handles unseen neurons effectively because: (1) the neural encoder unifies the latent space dimensionality, and (2) based on PNBA, training across multiple subjects with diverse neural populations encourages extraction of behaviorally-relevant features rather than memorizing individual neuron characteristics, enabling generalization to completely new neural populations.
> We will incorporate these details into the main text.
>
> ```Q5. Suggested SwapVAE```
>
> **A5**: Thank you for suggesting this reference. While both approaches address neural representations, fundamental differences exist between SwapVAE and our PNBA framework. DropVAE operates within a single modality (neural spikes), using data augmentation and swap operations predicated on within-trial similarity, without directly incorporating behavioral constraints. In contrast, PNBA draws inspiration from CLIP's multi-modal paradigm, explicitly aligning neural and behavioral representations through generative constraints. This cross-modal approach directly establishes neural-behavioral associations, enabling zero-shot generalization to unseen subjects, as detailed in **A4**. We will add this in our related work.

---

### Official Review · Reviewer_a8zJ · 2025-03-20

**Overall Recommendation:** 3

**Summary:**

This paper introduces PNBA, a framework leveraging probabilistic modeling to find robust preserved representations across different scales of neural variability: trials, sessions, and subjects. The method is evaluated on three datasets spanning M1, PMd and V1 of primates and mice, showing zero-shot preserved representations across cortices and species.

**Claims And Evidence:**

The paper claims to obtain robust neural-behavioral representation alignment within multiple cortical regions and from different species. This claim is convincingly supported by the Pearson correlation coefficients between neural and behavioral representations throughout the paper.

The second claim the paper made is on preserved neural representations through zero-shot validation with practical applications in zero-shot behavior decoding. However, it was not clear how this zero-shot generalization was achieved, given that each session/subject has varying number of neurons, making the application of the same encoder on the unseen session/subject without any finetuning impossible.

**Essential References Not Discussed:**

N/A

**Experimental Designs Or Analyses:**

The experimental designs and analyses look good generally, although some details are missing (see Questions for Authors)

**Methods And Evaluation Criteria:**

The choice of datasets (Safaie et al., 202, Turishcheva et al., 2024) and evaluation metrics made sense for this problem.

**Other Comments Or Suggestions:**

Minor typo on line 381 and Figure 1c: "trails" instead of "trials"

**Other Strengths And Weaknesses:**

Strengths:
* The paper tackles an important problem in neuroscience, which is finding a robust representation amidst neural variability across trials, sessions and subjects.
* Claims are supported with good experimental results.
* The paper is well written and easy to follow.

Weaknesses:
* It is unclear how the model handles varying neural population sizes for zero-shot behavior decoding in unseen sessions/subjects.
* The paper provides results of behavior decoding on V1 dataset but not M1 and PMd datasets.
* The related alignment method in Safaie et al, 2023 was mentioned (Figure 1) but was not compared with the proposed method in Table 1.
* Some details are missing that make it difficult to evaluate the soundness of the methods (see Questions for Authors)

**Questions For Authors:**

1. Figure 3a: Was the histogram constructed using aggregated samples from two mice? I'm wondering what the histograms on each individual mouse look like.
2. Figure 3a: What are the samples used to construct the histogram? Are the samples Nx1 vector at each timestep or NxT matrix of a trial/session? How are the normalized representation values computed from these vectors/matrices? How are trials of varying lengths and population sizes handled?
3. Figure 3b: Why is the correlation matrix not symmetric?
4. Line 247: How are the 4 pairs chosen from the 8 mice? Shouldn't there be 8 choose 2 total number of pairs?
5. Figure 5a: Is mean R calculated across all possible pairs of trials regardless of which behavior conditions that trial belongs to? How to handle the problem that each trial can be of different lengths? What is the standard deviation calculated over?
6. Figure 5b and 5c: Similar question to the above. How are the mean R and standard deviation calculated for across-session and across-subject?
7. Section 4.5: How can the model be applied zero-shot for behavior decoding, given that the input to the model should be of fixed size while the number of neurons and time steps can vary across subjects? Could the authors provide the detailed step-by-step procedure of training and inference of the model?
8. Section 4.5: Could the authors also provide results of behavior decoding on the primate datasets?

**Relation To Broader Scientific Literature:**

The paper made contributions toward representation learning in neuroscience, tackling the problem of identifying robust cross-modality representations across sessions and animals performing the same behavior tasks. The method has implications for calibration-free brain-computer interfaces with potential for motor function restoration.

**Theoretical Claims:**

I have not carefully checked the correctness of proofs for Theorem 3.1 which was in the Appendix.

---

> ### Author Rebuttal · Authors · 2025-03-28
>
> We appreciate the reviewer's detailed feedbacks! We address each point as follows:
>
> ```Q1. How does zero-shot generalization work with varying neuron counts across subjects?```
>
> **A1**: Our model uses convolutional and pooling layers to **standardize input activities of varying counts into a fixed-size latent space(detailed in SI Line 1088-1094)**. Based on **PNBA**, **training on multi-subject** further forces the model to learn behavior-relevant neural representations rather than individual-specific neuron characteristics, enabling cross-individual generalization without any fine-tuning on zero-shot subject.
>
> ```Q2. Safaie et al, 2023 was not compared in Table 1.```
>
> **A2**: Table 1 excludes Safaie et al. because they focus on neural representation alignment requiring individual-specific optimization, whereas we evaluate cross-modal correlations on unseen subjects without fine-tuning. A direct comparison would be unfair to theirs.
>
> ```Q3. Figure 3a: Was histogram from aggregated data of two mice?What do individual mouse histograms look like?```
>
> **A3**: Yes, the histogram aggregates data from 2 mice viewed identical stimuli. Individual mouse histograms show the same patterns for stimulus representation, with subtle peak variations in neural representation due to individual differences.
>
> ```Q4. Figure 3a:What samples construct the histogram?How are normalized values computed and varying trial lengths/population sizes handled?```
>
> **A4**: The histogram uses latent representations z (d×T) from each trial, reshaped into one-dimensional vectors and z-score normalized. Neuron count variations are addressed by pooling neural features into a fixed-dimensional latent space (SI Line 1088-1094). Variable trial lengths are handled through sliding window (t=16) processing (SI Line 1067). All vectors across trials are then concatenated into a one-dimensional array, whose frequency distribution forms the histogram.
>
> ```Q5. Figure 3b: Why is the correlation matrix not symmetric?```
>
> **A5**: The correlation matrix is asymmetric due to trial asymmetry. Specifically, corr_matrix[i,j] measures correlation between neural activity of trial i and visual stimulus of trial j, noted as pair (i,j), while corr_matrix[j,i] measures pair (j,i), a different combination, naturally yielding different values. Similarly in Figure 6c, pairs (i,j) and (j,i) represent different trial combinations.
>
> ```Q6. Line 247: How are the 4 pairs chosen from the 8 mice?Shouldn't there be 8 choose 2 total number of pairs?```
>
> **A6**: We apologize for the confusion. Our experiment included 5 pairs of mice (10 mice total), with each pair viewing identical stimuli. 4 pairs (8 mice) were used for training, while the remaining pair (2 mice) served as an independent test set to evaluate zero-shot performance (see SI Table 4). We will refine this in the revised version.
>
> ```Q7.  Figure 5a: is mean R calculated across all trial pairs regardless of behavior condition?How are different trial lengths handled?What is the standard deviation calculated over?```
>
> **A7**: No, we calculate mean R only between trials with matching behaviors, following Safaie et al, 2023. For trials with different lengths (d×T1 vs d×T2), e.g. T2>T1, we downsample T2 by uniformly discarding timepoints. This length handling only applies to monkey data, as all V1 trials have consistent lengths.
>
> Standard deviation is calculated across all possible same-behavior trial pairs.
>
> We will add these clarifications in our revision.
>
> ```Q8. Figure 5b and 5c: Similar question to the above.How are the mean R and standard deviation calculated for across-session and across-subject?```
>
> **A8**: We use identical strategy as **A7**, but comparing trials from different sessions or different subjects while maintaining matched behavioral conditions.
>
> ```Q9. Section 4.5: How can the model handle zero-shot behavior decoding with varying neuron counts and time steps?What's the detailed training and inference procedure?```
>
> **A9**: Neural representations have the same spatial dimensions across subjects (**A1**). V1 trials have consistent lengths, but we can use sliding windows to handle varying lengths (SI Line 1067).
>
> **The full procedure:(1) train PNBA on training subjects (2) generate representations via the frozen neural encoder, validate preservation (3) train V1 BCI decoders using training subjects' representations and behavior (4) apply frozen neural encoder and BCI decoder to unseen subjects.**
>
> ```Q10. Section 4.5: Could the authors also provide results of behavior decoding on the primate datasets?```
>
> **A10**: Our primate decoding results on unseen animals: M1 achieved R²=0.78; PMd achieved R²=0.71. We note that these results are expected as PNBA aligns neural-behavioral representations for these datasets. To avoid circular reasoning, we showed independent movement decoding using V1 data in Section 4.5, where the neural encoder was only trained with stimulus in PNBA.
>
> **Others**: We have fixed all typos.

---

### Decision · Program_Chairs · 2025-05-01

**Decision:**

Accept (poster)

**Comment:**

This paper proposes an approach to joint neural-behavioral alignment for latent representations. Specifically, the authors propose an objective that combines distribution matching (neural and behavioral data should map to similar latent distributions) with bounds on the joint and conditional probabilities of both data sources (similar to a symmetric information bottleneck, in which the latent must preserve information for decoding both modes, preventing mode collapse). The authors demonstrate that this objective prevents certain pathologies of distribution matching, and, in a series of experiments using multi-subject, multi-region recording data, demonstrate that representations inferred from behavior correlate well with those inferred from neural data. They then apply these same learned encoders to new subjects, finding strong zero-shot correlation in the same tests.

Reviewers agreed that this is an important problem in neuroscience and that the approach is novel. They also agreed that the model showed good performance again the selected benchmark methods. Concerns centered on a few themes:
1. Reviewers were confused by the method used to train on disparate numbers of neurons and different neuron identities across sessions. Authors clarified ll.1088-94 that this was done by a trainable session-specific projection at the encoder and decoder ends of the network. Thus, generalization to held-out subjects and sessions would still appear to require training these linear mappings, somewhat vitiating claims of zero-shot transfer.
2. Reviewers were also somewhat confused about the relationship between the proposed method and numerous other alignment methods in the literature. Authors clarified that while other methods focus on aligning, e.g., neural data across individuals for the same sensory inputs, their focus is on aligning latent representations inferred from jointly recorded neural and behavioral data. Authors would do well to clarify further in a revised manuscript.
3. There was some question as to how these methods would generalize to realistic BCI settings with significant constraints on computation and streaming.

In all, reviewers found the experiments strong and method innovative, suggesting a valuable contribution to the alignment literature.